# Astrocytic metabolic control of orexinergic activity in the lateral hypothalamus regulates sleep and wake architecture

Alice Braga[1,4], Martina Chiacchiaretta ®[1,4] ✉, Luc Pellerin ®[2], Dong Kong ®[1,3] & Philip G. Haydon ®[1] ✉

Neuronal activity undergoes significant changes during vigilance states, accompanied by an accommodation of energy demands. While the astrocyte-neuron lactate shuttle has shown that lactate is the primary energy substrate for sustaining neuronal activity in multiple brain regions, its role in regulating sleep/wake architecture is not fully understood. Here we investigated the involvement of astrocytic lactate supply in maintaining consolidated wakefulness by downregulating, in a cell-specific manner, the expression of monocarboxylate transporters (MCTs) in the lateral hypothalamus of transgenic mice. Our results demonstrate that reduced expression of MCT4 in astrocytes disrupts lactate supply to wake-promoting orexin neurons, impairing wakefulness stability. Additionally, we show that MCT2-mediated lactate uptake is necessary for maintaining tonic firing of orexin neurons and stabilizing wakefulness. Our findings provide both in vivo and in vitro evidence supporting the role of astrocyte-to-orexinergic neuron lactate shuttle in regulating proper sleep/wake stability.

Early neuroenergetics studies proposed glucose as the key energy source for neuronal energetic demand, with the classical concept that lactate represents a mere by-product of glycolysis; however, recent evidence highlighted a more essential role for this metabolite[1]. Lactate transport in the brain is facilitated by monocarboxylate transporters (MCTs)[2]: MCT1 expressed by endothelial cells[3,4] oligodendrocytes[5] and astrocytes[6,7]; MCT2 found primarily in neurons[8] and MCT4 is mainly expressed by astrocytes and microglia[9–11]. Disrupting the lactate shuttle via pharmacological blockage of MCTs or antisense oligonucleotides has been shown to impair synaptic function in the hippocampus[12], memory processing[11,13–15], and motor function[10], supporting the importance of these transporters in several physiological behaviors. In vivo studies showed that cerebral lactate in the cortex fluctuates depending on the animal's vigilant state[16,17], with a consistent increase upon

wakefulness followed by a reduction during sleep[18,19]. Interestingly, the activity of wake-promoting orexin neurons in the lateral hypothalamus (LH), a brain area involved in sleep/wake cycle control, increases during wakefulness, and decreases during sleep. While exploratory in vitro studies showed the excitatory effect of lactate on orexinergic activity[20] and their preference for lactate as an energy source[21], several key aspects remain unexplored. These include identifying the specific cellular origin and transporters that give rise to lactate during wakefulness in the LH. Additionally, determining the relevance of lactate through the sleep/wake cycle as an energy source under physiological conditions of glucose availability.

Here, we selectively reduced the expression of astrocytic MCT4 or MCT1 and orexinergic MCT2 in the LH to explore the contribution of each MCT isoform in regulating orexinergic activity and the impact on sleep/wake cycle. By using both in vivo electroencephalographic

[1]Department of Neuroscience, Tufts University School of Medicine, Boston, MA 02111, USA. [2]Inserm U1313, University and CHU of Poitiers, 86021 Poitiers, France. [3]Division of Endocrinology, Department of Pediatrics, F.M. Kirby Neurobiology Center, Boston Children's Hospital and Harvard Medical School, Boston, MA 02115, USA. [4]These authors contributed equally: Alice Braga, Martina Chiacchiaretta. ✉e-mail: martina.chiacchiaretta16@gmail.com; philip.haydon@tufts.edu

(EEG)/electromyographic (EMG) recordings coupled with real-time lactate biosensor measurements and in vitro electrophysiology, we showed that astrocytes shuttle lactate into the extracellular space mainly via MCT4 upon wakefulness and such mechanism supports orexinergic neurons firing, ultimately promoting wakefulness. We also showed the role of orexinergic MCT2 in promoting their activity and maintaining wakefulness via lactate uptake. Taken together, our data show a tightly coupled astrocytic-neuronal metabolic communication in maintaining a proper sleep/wake cycle. This is of particular relevance given the significant relationship between impairment of sleep/wake architecture and a wide range of disorders[22,23].

## Results

### Blockage of MCTs or glycogen mobilization in the LH decreases wakefulness in the dark phase

The astrocyte-neuron lactate shuttle (ANLS) hypothesis postulates that astrocytes export lactate into the extracellular milieu as a metabolic substrate for neuronal consumption[24]; data from our previous work showed that the astrocytic connexin-43 network within the LH is a possible route for trafficking lactate that is critical for maintaining consolidated wakefulness[21]. Nevertheless, the exact role of the ANLS within the LH in regulating sleep and wakefulness is yet to be described.

To begin to test the importance of astrocytes as a lactate source in the LH, we pharmacologically inhibited glycogenolysis, since glycogen is a major source of lactate and is exclusively stored in astrocytes[25,26]. To investigate whether glycogen-derived lactate in the LH is required for sustained wakefulness we used bilateral cannula connected to osmotic mini-pumps, together with EEG/EMG recordings (Fig. 1a), to deliver aCSF followed by the inhibitor of glycogen phosphorylase 1,4-dideoxy-1,4-imino-D-arabinitol (DAB; 10 mM) (Fig. 1b). Blockage of glycogenolysis resulted in a 14.56% reduction of time spent in wakefulness, mirrored by a 14.26% increase in time spent in non-rapid eye movement (NREM) during the first hours of the dark phase (ZT12-18) (Fig. 1b, c and Supplementary Fig. 1). This window of sensitivity (ZT 12-18) is consistent with previous studies that have highlighted this period of the dark phase as being particularly sensitive to metabolic disruption in the LH[21]. To test whether lactate was able to restore prolonged wakefulness, we then delivered DAB in combination with L-lactate (DAB 10 mM; L-lactate 5 mM). Treatment with L-lactate was able to reverse the effect of blockage of glycogenolysis, with 16.54% increase in time spent in wakefulness and 16.33% reduced time in NREM during the first hours of the dark phase (ZT12-18) (Fig. 1b, c and Supplementary Fig. 1). No significant differences in number or duration of wake or sleep events were associated with these changes in overall wakefulness architecture (Fig. 1d, e). These data suggest that lactate produced through astrocytic glycogen mobilization promotes wakefulness in the dark phase.

Next, we asked whether MCTs, the major transporters of lactate in the brain, are required for sustained wakefulness. Delivery of the non-selective MCT inhibitor alpha-cyano-4-hydroxycinnamate (4-CIN; 1.5 mM) (Fig. 1f) revealed a significant effect during the initial 6 h of the dark phase, resulting in an 11.96% reduction in time spent in wakefulness (ZT12-18) (Fig. 1g and Supplementary Fig. 1). This was accompanied by an increase in time spent in both NREM (11.57% increase) and rapid eye movement (REM) sleep (0.21% increase) (Fig. 1g and Supplementary Fig. 1). In addition, we observed an increase in the number of NREM episodes, as well as in the number and duration of REM episodes during ZT12-18 (Fig. 1h, i). We also detected a 3.51% increase in the time spent in wakefulness during the light phase (Fig. 1g and Supplementary Fig. 1).

Collectively, these results support our previous observations[21] and suggest that the shuttling of lactate from astrocytes to neurons in the LH is necessary for promoting and maintaining prolonged wakefulness in mice during the dark phase.

### Astrocytic MCT4 is required for sustained wakefulness during the dark phase

To further investigate the source of extracellular lactate and its impact on wakefulness, we performed EEG/EMG recordings on mice with astrocyte-specific MCT knockdown. Astrocytes predominantly express two MCT isoforms, MCT4 and MCT1[6,27]. To discern whether either MCT isoforms contributes to the regulation of sleep-wake cycle within the LH, we employed mice with loxP sites flanking exons 3 to 5 of the *Mct4* gene (MCT4[f/f]) (Supplementary Fig. 2a) and exon 5 of the *Mct1* gene (MCT1[f/f])[28,29] (Supplementary Fig. 2f). MCT4[f/f] and MCT1[f/f] mice were injected with PhP.eB-GFAP(0.7)-eGFP-T2A-iCre (aMCT4 KD or aMCT1 KD) or the control virus PhP.eB-GFAP(0.7)-eGFP in the LH (Fig. 2a and Supplementary Fig. 4b). We first assessed astrocyte-specific selectivity by injecting Ai14-reporter mice in the LH with adeno-associated viruses (AAVs) encoding GFAP(0.7)-eGFP-T2A-iCre. Quantification of double positive cells expressing both the endogenous Td-Tomato, expressed upon Cre recombination, and the astrocytic (GFAP) or orexin (Orexin-A) markers showed high selectivity of the GFAP(0.7) promoter for astrocytes (Supplementary Fig. 2b). We next confirmed the excision of the floxed exons upon Cre recombination (Supplementary Fig. 2a, f), followed by an assessment of MCT4 and MCT1 protein expression. Six weeks post viral injection, there were no alterations in MCT4 protein expression (Supplementary Fig. 2c), nor changes in sleep/wake architecture (Supplementary Fig. 2d, e). However, twelve weeks post viral injection, MCT4 protein expression was reduced by 48% compared to the control group and sleep/wake architecture was perturbed (Supplementary Fig. 3a, Fig. 2b). At this timepoint aMCT4 KD mice showed a 12.40% reduction in wakefulness during ZT12-18 (Fig. 2c), accompanied by an 11.73% increase in NREM sleep (Fig. 2c and Supplementary Fig. 3). aMCT4 KD mice also showed an increased number of shorter wakefulness events and increased number of NREM events during ZT 12-18 (Fig. 2d, e). We asked whether in vivo delivery of lactate to the LH would rescue the wakefulness instability induced in aMCT4 KD mice by using bilateral cannula connected to osmotic mini-pumps and EEG/EMG recording (Fig. 2f). Compared to aCSF, the delivery of lactate to aMCT4 KD mice increased wakefulness during the dark phase (Fig. 2g). We observed a 15.76% increase in duration of time spent in wakefulness mirrored by a 15.15% reduction of time spent in NREM during the first hours of the dark phase (ZT12-18; Fig. 2h and Supplementary Fig. 3). We also observed an 11.43% increase of time spent in wakefulness during the second part of the dark phase (ZT18-24) (Fig. 2h and Supplementary Fig. 3). Finally, lactate delivery also resulted in reduced number of NREM events during the first hours of the dark (ZT 12-18) (Fig. 2i). No changes in the duration of events were observed (Fig. 2j).

We then tested the role of astrocytic MCT1, six weeks after viral injections we observed an increased expression of MCT1 protein (Supplementary Fig. 2g) accompanied by no changes in sleep/wake architecture (Supplementary Fig. 2h, i). Given the highly abundant distribution of MCT1 among CNS cells, such as oligodendrocytes[5] or endothelial cells[4] we hypothesized an acute compensatory over-expression. Twelve weeks after viral injections, MCT1 expression was reduced by 12% compared to the control group (Supplementary Fig. 4a), however, we did not observe any change in the percentage of time the mice spent awake and in NREM and REM sleep (Supplementary Fig. 4c, d). aMCT1 KD mice showed only an increase in the number of wake events (Supplementary Fig. 4e). We also observed a small increased number of wake events with shorter duration during ZT0-12 (Supplementary Fig. 4f). These data indicate that MCT1 has little to no role in supporting the maintenance of consolidated wakefulness during the dark phase.

Together, these results suggest that astrocytic MCT4 supports wakefulness during ZT12-18, consistent with the transport of lactate into the extracellular milieu by MCT4 given that lactate can rescue the aMCT4 KD phenotype.

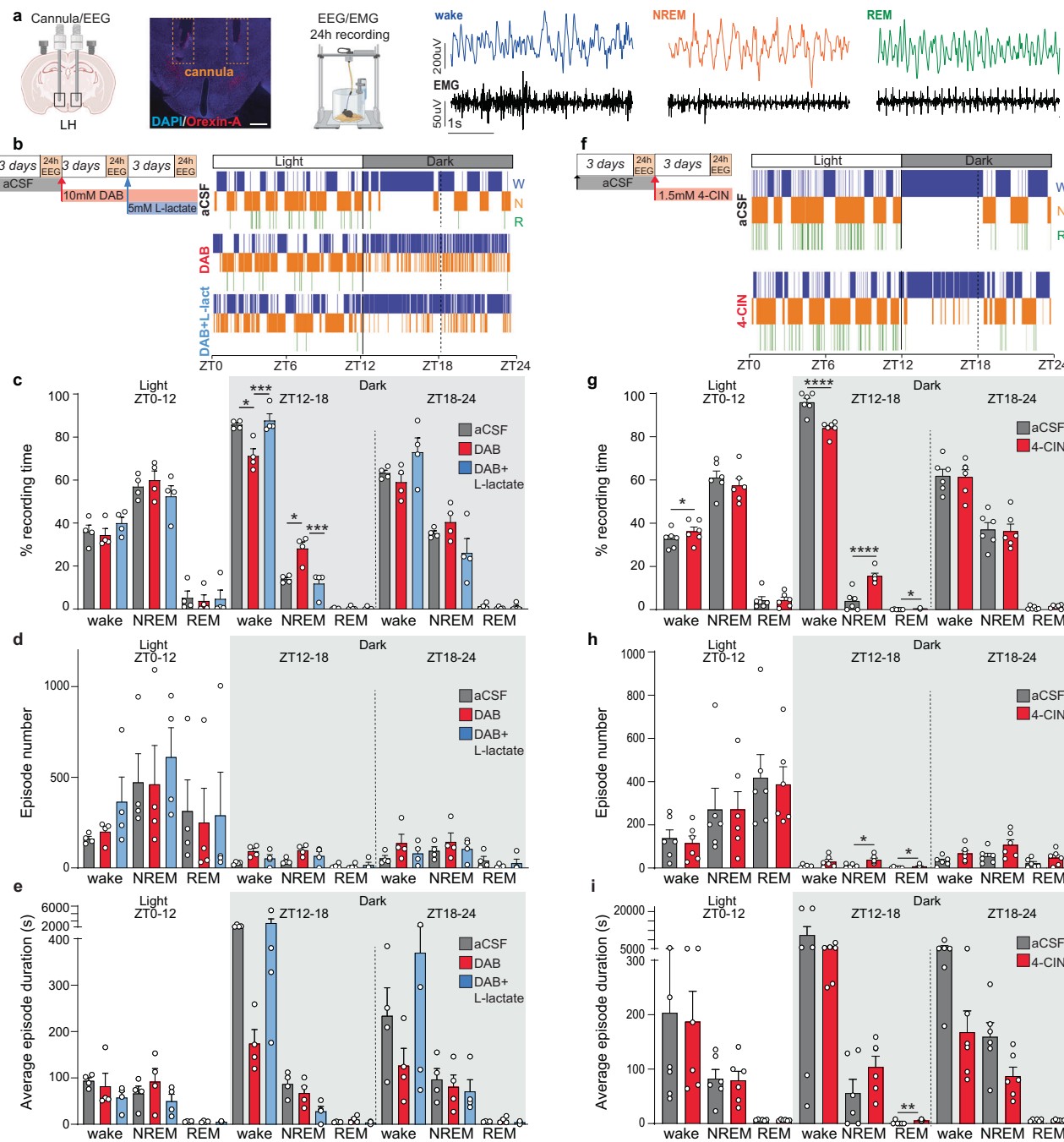

**Fig. 1 | Blockage of glycogenolysis or MCTs in the LH reduces wakefulness during the dark phase. a** (Left to right) Cartoon representing implantation of bilateral cannula in the LH for drug delivery coupled with EEG/EMG recording. Immunofluorescence micrograph showing placement of the cannula (orange squares) near to Orexin-A positive cells (red). Scale bar = 100 μm. Schematic representation of EEG/EMG recording setting and representative traces for EEG and EMG during wakefulness, NREM and REM. Experimental design and timeline for DAB and DAB + L-lactate infusion (**b**) and 4-CIN infusion (**f**). Representative hypnograms of the same WT mouse in the different conditions during 24 h EEG recordings (W – wake; N – NREM and R – REM). Quantification of the averaged percentage of time spent in wake, NREM, or REM before and after DAB, DAB + L-lactate (**c**) or 4-CIN treatment (**g**) during the 24 h recording phase. Average of the number of wake, NREM, and REM episodes during the 24 h recording phase before

and after DAB, DAB + L-lactate (**d**) or 4-CIN (**h**). Average duration of wake, NREM, and REM episodes during the 24 h recording phase before and after DAB, DAB + L-lactate (**e**) or 4-CIN infusion (**i**). (**c**, **d** and **e**: n = 4, Repeated-measure ANOVA followed by Tukey's post hoc test; ZT12-18 %wake *p = 0.033, ***p = 0.0009, %NREM *p = 0.030, ***p = 0.0006; aCSF infusion in gray, DAB infusion in red, DAB + L-lactate infusion in blue; **g**, **h** and **i**: n = 6, two-tailed paired t test; ZT0-12 %wake *p = 0.039, ZT12-18 %wake and %NREM: ****p < 0.0001, %REM *p = 0.049, episode number REM *p = 0.022, episode number NREM *p = 0.042, episode duration REM *p = 0.009; aCSF infusion in gray, 4-CIN infusion in red). ZT= Zeitgeber Time; EEG= electroencephalogram; EMG=electromyography; DAB = 1,4-dideoxy-1,4-imino-d-arabinitol; 4-CIN=alpha-cyano-4-hydroxycinnamate. Pooled data are shown as mean ± SEM. Source data are provided as a Source Data file.

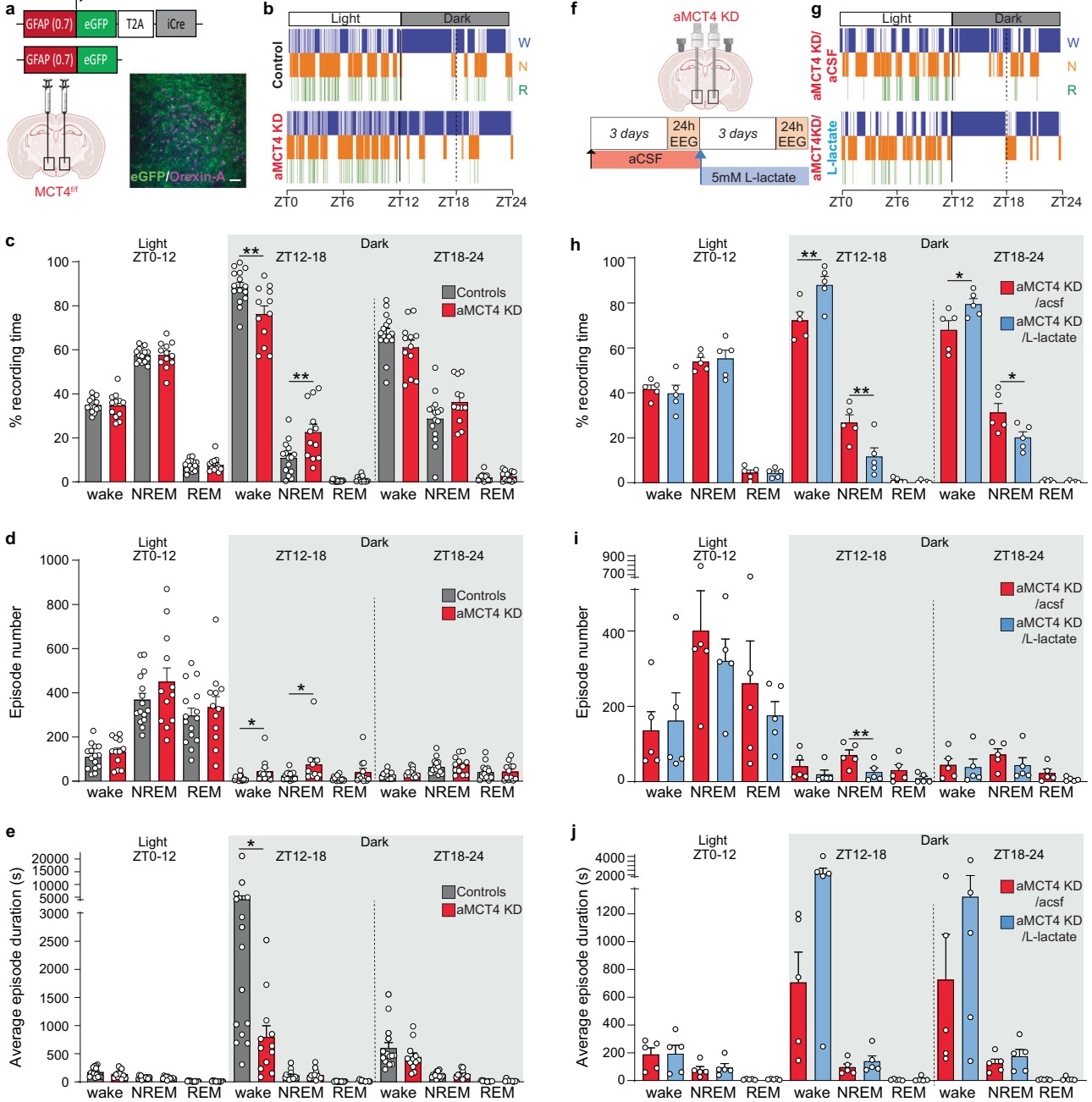

**Fig. 2 | Deletion of astrocytic MCT4 induces excessive sleepiness during the dark phase and is rescued by exogenous lactate. a** Schematic of stereotaxic injections of either Cre-encoding PhP.eB virus (aMCT4 KD mice) or control PhP.eB virus (control mice) in the LH of MCT4$^{f/f}$ mice and immunofluorescence micrograph showing eGFP expression (green) near Orexin-A (magenta) positive cells 12 weeks after virus injection. Scale bar = 100 μm. Representative hypnograms of Control mouse and aMCT4 KD mouse during 24 h EEG recordings (**b**) or same aMCT4 KD before and after L-lactate infusion (**g**) (W – wake; N – NREM and R – REM). Quantification of the average percentage of time spent in wake, NREM or REM sleep during the 24 h recording phase in aMCT4 KD mice compared to control mice (**c**) or in aMCT4 KD mice before and lactate infusion (**h**). Average of the number of wake, NREM and REM episodes during the 24 h recording phase in aMCT4 KD mice

compared to control mice (**d**) or in aMCT4 KD mice before and after lactate infusion (**i**). Average duration of the wake, NREM and REM episodes during the 24 h recording phase in aMCT4 KD mice compared to control mice (**e**) or in aMCT4 KD mice before and after lactate infusion (**j**) (**c, d, e**: red, aMCT4 KD $n = 12$; gray, controls $n = 15$, two-tailed unpaired $t$ test; ZT12-18: %wake **$p = 0.005$, %NREM $p = 0.006$; episode number wake *$p = 0.033$, episode number NREM *$p = 0.040$, episode duration wake *$p = 0.048$; **h, i, j**: $n = 5$, two-tailed paired $t$ test; ZT12-18 %wake and % NREM **$p = 0.003$; ZT18-24 %wake *$p = 0.039$, %NREM *$p = 0.043$; ZT12-18 episode number NREM **$p = 0.005$). ZT = Zeitgeber Time. Pooled data are shown as mean ± SEM. **f** Experimental design and timeline for L-lactate infusion; 24 h EEG recordings were analyzed over time in the same aMCT4 KD mouse. Source data are provided as a Source Data file.

## Orexinergic MCT2 is required for consolidated wakefulness

To test the role of orexinergic MCT2, the major monocarboxylate isoform expressed in neurons[8] in the control of wakefulness, we crossed Orexin-IRES-Cre mice with mice that have loxP sites flanking exons 4 and 5 of MCT2 (MCT2$^{f/f}$)[28] to obtain orexin neuron-specific

knockdown mice (oMCT2 KD mice) (Fig. 3a and Supplementary Fig. 5a). We first investigated whether early modulation of MCT2 could affect orexin neurons via immunohistochemistry techniques, and we did not observe any changes in the number of Orexin-A positive cells in oMCT2 KD mice compared to control mice (Supplementary Fig. 5b).

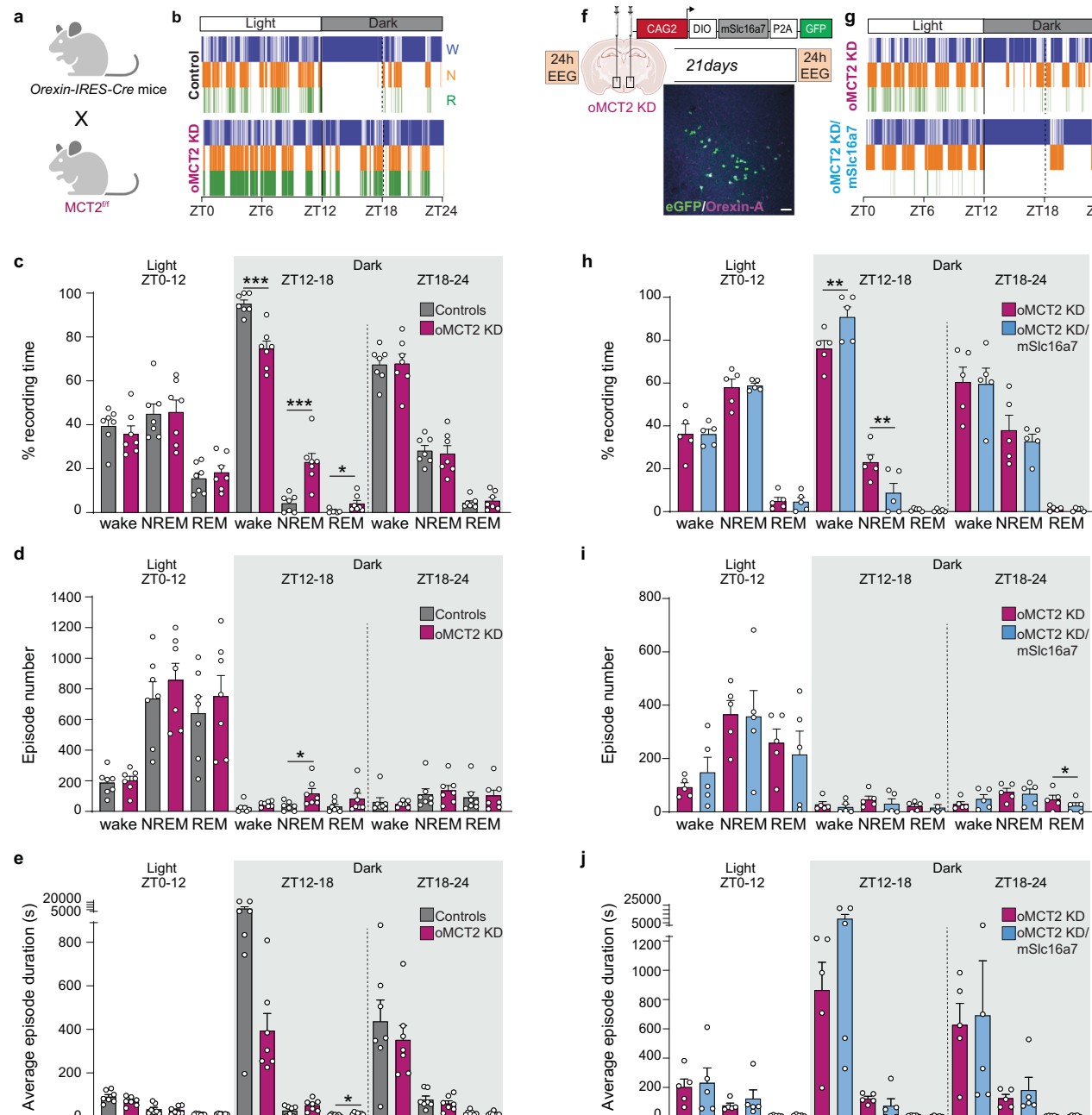

**Fig. 3 | Deletion of orexinergic MCT2 induces excessive sleepiness during the dark phase. a** Schematic representation showing breeding strategies for deletion of MCT2 in orexin neurons: Orexin-IRES-Cre mice were crossed with MCT2$^{f/f}$ (oMCT2 KD), Cre-negative littermates were used as controls. **b, g** Representative hypnograms during 24 h EEG recordings of Control mouse and oMCT2 KD mouse (**b**) or same oMCT2 KD mouse before and after MCT2 re-expression (W – wake, N – NREM, R – REM). Quantification of averaged percentage of time spent in wake, NREM, or REM sleep during the 24 h recording phase in oMCT2 KD mice compared to control mice (**c**) or before and after MCT2 re-expression in oMCT2 KD (**h**). Average of the number of wakefulness, NREM, and REM episodes during the 24 h recording phase in oMCT2 KD mice compared to control mice (**d**) or before and after MCT2 re-expression in oMCT2 KD (**i**). Average duration of the wake, NREM, and REM episodes during the 24 h recording phase in oMCT2 KD mice compared to control mice (**e**) or before and after MCT2 re-expression in oMCT2 KD (**j**). **c**, **d**, **e**: controls $n = 7$, oMCT2 KD $n = 7$, two-tailed unpaired $t$ test; ZT12-18 %wake ***$p = 0.0002$, %NREM ***$p = 0.0008$, %REM *$p = 0.035$; episode number NREM *$p = 0.029$; episode duration REM *$p = 0.044$; **h, i, j**: $n = 5$, two-tailed paired $t$ test; ZT12-18 %wake **$p = 0.009$, %NREM **$p$-0.008, ZT18-14 episode number REM *$p = 0.029$. ZT= Zeitgeber Time. Pooled data are shown as mean ± SEM. **f** Experimental design and timeline of local LH injection of PhP.eB virus expressing mScla16a7 in the presence of CRE (DIO) in oMCT2 KD mice. Twenty-four hours EEG recordings were analyzed over time in the same aMCT2 KD mouse before virus injection and 21 days after injection. Scale bar = 100 μm. Source data are provided as a Source Data file.

Twenty-four hours EEG/EMG recording showed that oMCT2 KD mice were unable to sustain consolidated wakefulness (Fig. 3b), characterized by a 20.42% reduction in time spent in wakefulness, an 18.80% increase in time spent in NREM and a 3.46% increase in REM during the first hours of the dark phase (Fig. 3c and Supplementary Fig. 5). oMCT2

KD mice also showed an increased number of NREM events and longer REM episodes during the dark period (Fig. 3d, e).

We then aimed at rescuing the wakefulness phenotype observed in oMCT2 KD mice by viral re-expression of MCT2 in oMCT2 KD mice. Adeno-associated virus (AAV) encoding DIO-mSlc16a7-P2A-GFP was

injected in the LH of oMCT2 KD mice to achieve Cre-mediated and orexin neuron-specific expression of MCT2 (Fig. 3f). Twenty-four hours EEG/EMG was recorded prior to and 21 days after viral injection. Re-expression of MCT2 rescued consolidated wakefulness during the first hours of the dark (Fig. 3g), with a 14.77% increase in time spent in wakefulness, accompanied by a 14.32% reduction of time in NREM (Fig. 3h and Supplementary Fig. 5) compared to pre-expression levels. Additionally, MCT2 re-expressed mice also showed a reduced number of REM episodes during the last hours of the dark phase (Fig. 3i), with no changes in the duration of events (Fig. 3j). These data support the notion that orexinergic neuronal MCT2 is required for sustained wakefulness during the dark period.

### Lactate shuttling via MCT4 and MCT2 is required for sustained orexin neuron activity

Orexin neurons in the LH are known to promote wakefulness by increasing their activity during the transition from sleep to wake[18,30]. To evaluate whether reduced wakefulness observed in aMCT4 KD and oMCT2 KD mice was correlated with changes in orexinergic activity we performed whole-cell recordings from orexin neurons in the hypothalamic brain slices (Fig. 4a, b). Throughout all the recordings the slices were perfused with ACSF containing a physiological concentration of glucose (2.5 mM)[31]. Orexin neurons were discriminated from co-located melanin-concentrating (MCH) neurons present in the same area by their distinct firing patterns. Post-hoc immunostaining against orexin A (Orexin-A), of biocytin-filled neurons confirmed that they were Orexin-A-positive (Supplementary Fig. 6a).

We found a profound decrease in orexinergic activity in whole-cell patch clamp mode in aMCT4 KD accompanied by a more hyperpolarized resting membrane potential compared to control mice (Fig. 4b, c, d). Accordingly, neurons from aMCT4 KD also exhibited a reduction in input resistance, suggesting that the decreased activity was mediated by an increase in ion conductance in these cells (Fig. 4e). To control for the possibility that the dilution of the intracellular compartment by whole-cell dialysis altered the metabolic state of the cell we repeated the same measurements in cell-attached mode. These recordings confirmed a reduction in spontaneous firing rate in aMCT4 KD to a similar extent to what was observed in whole-cell mode (Supplementary Fig. 6b, c).

We then tested whether deletion of MCT2 impacted orexinergic activity. To verify the selective expression of Cre-recombinase in orexinergic neurons, we performed stereotaxic injections into the LH of Orexin-IRES-cre mice (Control and oMCT2 KD) with an AAV9-CAG-DIO-mCherry virus. Cre-dependent mCherry expression was evaluated through immunostaining for the orexin marker Orexin-A. We found that 94% ± 2.1% of mCherry expressing cells were Orexin-A positive and that 85.6% ± 4.9% of all orexinergic neurons were mCherry positive (Supplementary Fig. 7a, b). This faithful co-localization of mCherry and Orexin-A in the LH indicates that the expression of Cre-recombinase is specific for orexinergic neurons and that the majority of orexin neurons express Cre-recombinase. Whole-cell patch-clamp recordings revealed a decrease in orexinergic activity, a more hyperpolarized membrane potential in oMCT2 KD compared to control mice (Fig. 4f, g). A similar reduction in activity persisted in cell-attached mode (Supplementary Fig. 6b, c).

Given the role of MCTs in lactate trafficking[2], we speculated that the decrease in activity observed in orexin neurons was due to a decrease in lactate supply. Thus, we first tested whether bath perfusion of lactate could rescue orexinergic activity in aMCT4 KD mice. We found that bath application of 5 mM lactate induced a membrane potential depolarization of 11 ± 3.3 mV in aMCT4 KD, which was sufficient to increase the activity of orexinergic neurons (Fig. 4i, j), while in control cells extracellular lactate did not affect membrane potential and firing rate suggesting that the endogenous lactate level had reached a plateau at 2.5 mM glucose. (Fig. 4k and Supplementary

Fig. 8a, d). These data support the idea that the reduced activity observed in aMCT4 KD was due to an impairment in lactate shuttling. Interestingly, the recovering effect of lactate was abrogated by the extracellular perfusion of 500 μM 4-CIN, a broad-spectrum inhibitor of MCTs, suggesting that, to modulate orexinergic excitability, lactate needs to be transported rather than acting via an extracellular receptor (Fig. 4j).

To support the observation that maintenance of orexin neurons activity requires lactate internalization, we tested whether the decreased excitability and membrane hyperpolarization observed in oMCT2 KD mice could be restored by extracellular perfusion of 5 mM lactate. We found that lactate was unable to change orexinergic excitability, while tolbutamide, by inhibiting $K_{ATP}$ channel, depolarized and increased the firing rate on the same recorded cells (Fig. 4l, m). Previous studies have demonstrated that lactate acts through $K_{ATP}$ channel[32]. This channel senses the metabolic state of orexinergic neurons and mediates the hyperpolarization of these cells during an energy shortage[33]. Finally, to assess that the excitatory effect of lactate on orexinergic neurons was mediated by $K_{ATP}$ channel, brain slices from aMCT4 KD were perfused with 100 μM tolbutamide. Extracellular perfusion with tolbutamide increased orexinergic excitability in aMCT4 KD mice but not in control cells (Fig. 4n and Supplementary Fig. 8b, c).

These results suggest that lactate, shuttled by astrocytes through MCT4, needs to be internalized through MCT2 for sustaining orexinergic excitability and the activity of these neurons in oMCT2 KD and aMCT4 KD mice can be rescued by inhibiting $K_{ATP}$ channel activity.

### An astrocytic source and neuronal sink of lactate

According to our data, we propose that astrocytes release lactate into the extracellular space via MCT4 where it is taken up into orexin neurons by neuronal MCT2. To test this hypothesis, we determined the change in extracellular lactate concentration during the transition from sleep to wake in KD and control mice. We employed EEG recordings coupled with real-time lactate biosensor recording in the LH (Fig. 5a) that allowed us to measure the changes in lactate during the transition from sleep to wake events (Fig. 5b).

Analysis of the frequency distribution of lactate increases during wakefulness events in the first hours of the dark (ZT12-18) showed that aMCT4 KD mice were characterized by a significant decrease in extracellular lactate elevation during the transition to wakefulness compared to controls (Fig. 5c) displaying a median value of 0.07 mM in a MCT4 KD compared to 0.20 mM in control mice (Fig. 5d). These results are consistent with a reduction in release of lactate into the extracellular space. In contrast, deletion of neuronal MCT2 caused an increase in the wakefulness-dependent lactate transient consistent with an impaired uptake of lactate by orexinergic neurons (Fig. 5e). oMCT2 KD showed a median value corresponding to 0.21 mM compared to 0.15 in control mice (Fig. 5f). These data support the role of astrocytic MCT4 as the main release route to give rise to extracellular lactate surges during wakefulness in the LH, and that orexin neurons are a sink for lactate utilization via MCT2.

## Discussion

While there is increasing evidence supporting roles of the ANLS in various brain regions and functions, the cellular mechanism involved in lactate increase and its role as an energy source throughout the sleep/wake cycle needs to be elucidated. Here, we show that astrocytic MCT4, but not MCT1, supports prolonged wakefulness. We also demonstrate the necessity for lactate to be imported into orexinergic neurons via MCT2 to promote their activity and wakefulness. Finally, we show that despite the availability of physiological glucose, astrocytic lactate is necessary for sustaining orexinergic activity and consolidating wakefulness (Fig. 6a–c).

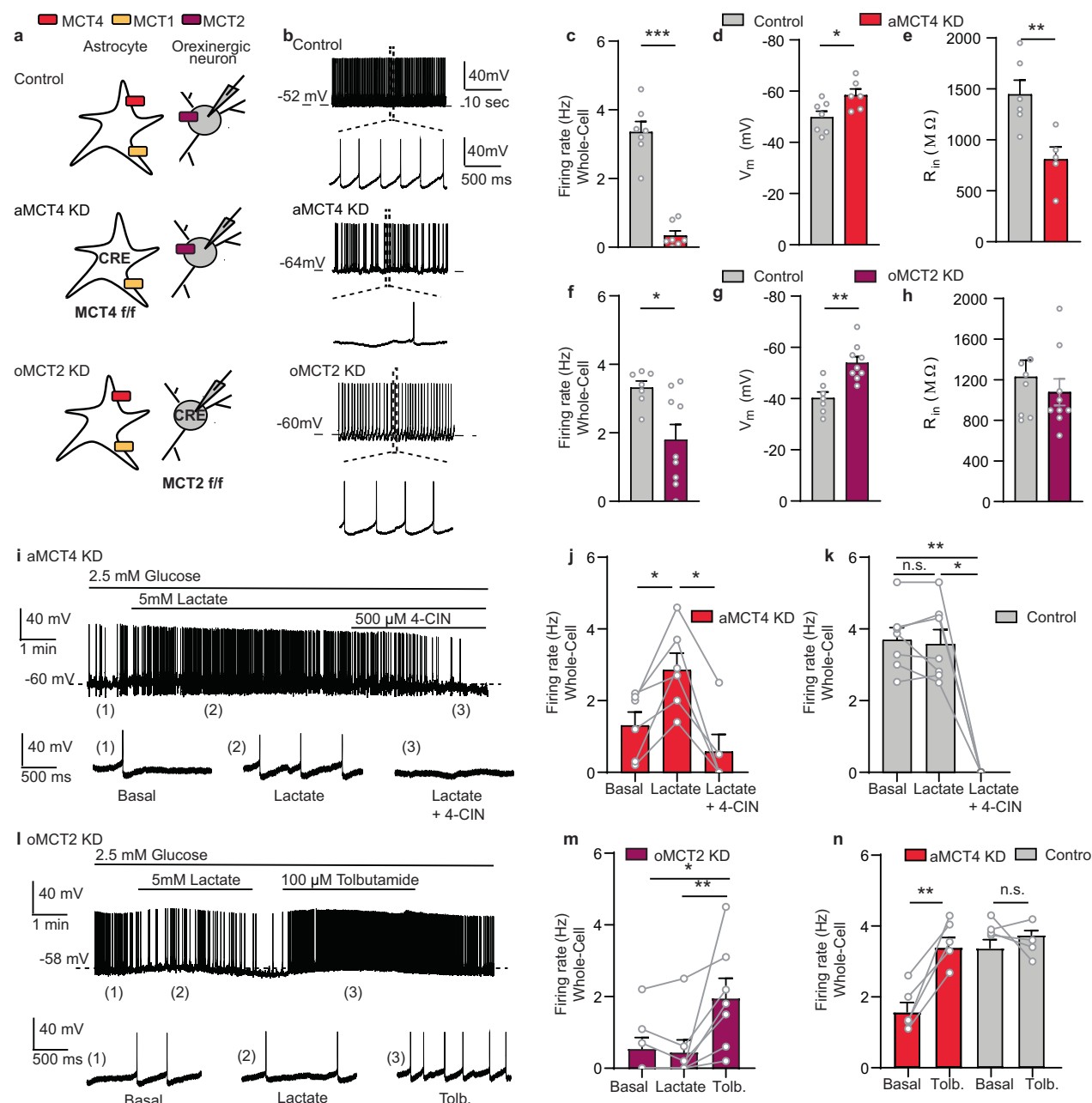

**Fig. 4 | Exogenous lactate restores spontaneous activity of orexinergic neurons from astrocytic MCT4 KD mice but not in orexinergic MCT2 KD mice.**
**a** Experimental design. Representative traces (**b**) and average firing rate (**c**, **f**) showing reduced firing in orexin neurons of aMCT4 KD (Controls and aMCT4 KD; $n = 7$ cells/4 mice; two-tailed Mann–Whitney test, ***$p = 0.0006$) and oMCT2 KD mice (Controls, $n = 7$ cells/4 mice; oMCT2 KD, $n = 9$ cells/4 mice; two-tailed Student $t$ test, *$p = 0.0131$). **d**, **g** Membrane voltage (Vm) in aMCT4 KD (Controls, $n = 7$ cells/4 mice; aMCT4 KD, $n = 6$ cells/3 mice; two-tailed Student $t$ test, *$p = 0.0252$) and oMCT2 KD mice (Control, $n = 7$ cells/4 mice and oMCT2 KD $n = 9$ cells/4mice; two-tailed Student $t$ test **$p = 0.0011$). **e**, **h** Average input resistance (Rin) in aMCT4 KD (Control, $n = 6$ cells/3 mice; aMCT4 KD, $n = 5$ cells/3 mice; two-tailed Student $t$ test, **$p = 0.0082$), and oMCT2 KD (Control, $n = 7$ cells/3 mice and oMCT2 KD $n = 9$ cells/4 mice; two-tailed Student $t$ test, $p = 0.28$). **i**–**k** Lactate depolarization of an aMCT4 KD neuron was suppressed by 4-CIN. Representative trace (**i**) and average firing rate

(**j**) ([Basal], $n = 6$ cells/4 mice; [Lactate], $n = 6$ cells/4 mice; [Lactate + 4-CIN], $n = 5$ cells/4 mice; Mixed-effects analysis followed by Tukey's multiple comparison test, *$p = 0.0107$, *$p = 0.0267$). Same experiment as in (**j**) but in Control mice (**k**), ([Basal], $n = 6$ cells/3 mice; [Lactate], $n = 6$ cells/3 mice; [Lactate + 4-CIN], $n = 4$ cells/3 mice; Mixed-effects analysis followed by Tukey's multiple comparison test, **$p = 0.008$, *$p = 0.0176$). No data points were excluded from the study. **l**, **m** Extracellular lactate had no effect on orexinergic activity from oMCT2 KD mice, while tolbutamide increased the activity. The bottom traces show an expanded timescale of the recording (**l**). Average firing (**m**) ($n = 7$ cells/5 mice; Friedman Test followed by Dunn's multiple comparisons test, *$p = 0.022$, **$p = 0.009$). **n** Tolbutamide increased orexinergic activity from aMCT4 KD ($n = 5$ cells/3 mice; two-Tailed paired Student's $t$ test, **$p = 0.0037$) but had no effect in Control mice ($n = 4$ cells/3 mice; two-Tailed paired Student's $t$ test; $p = 0.33$). Pooled data are expressed as Mean values ± SEM. Source data are provided as a Source Data file.

Our previous work demonstrated the importance of Cx43-mediated networks for sustaining orexin neurons activity and wakefulness[21]. We built upon this evidence to reveal the source and sink of lactate during vigilant states. How is lactate released into the extracellular space and how does it influence wakefulness across the

24 h? We first hypothesized that glycogen-derived lactate has a prominent role in fueling neuronal activity during sustained wakefulness. Astrocytes can produce lactate both from glycolysis and glycogen turnover[26,34,35]. Progressive reduction of glycogen granules is associated with progressively longer wake episodes[36]. We found that

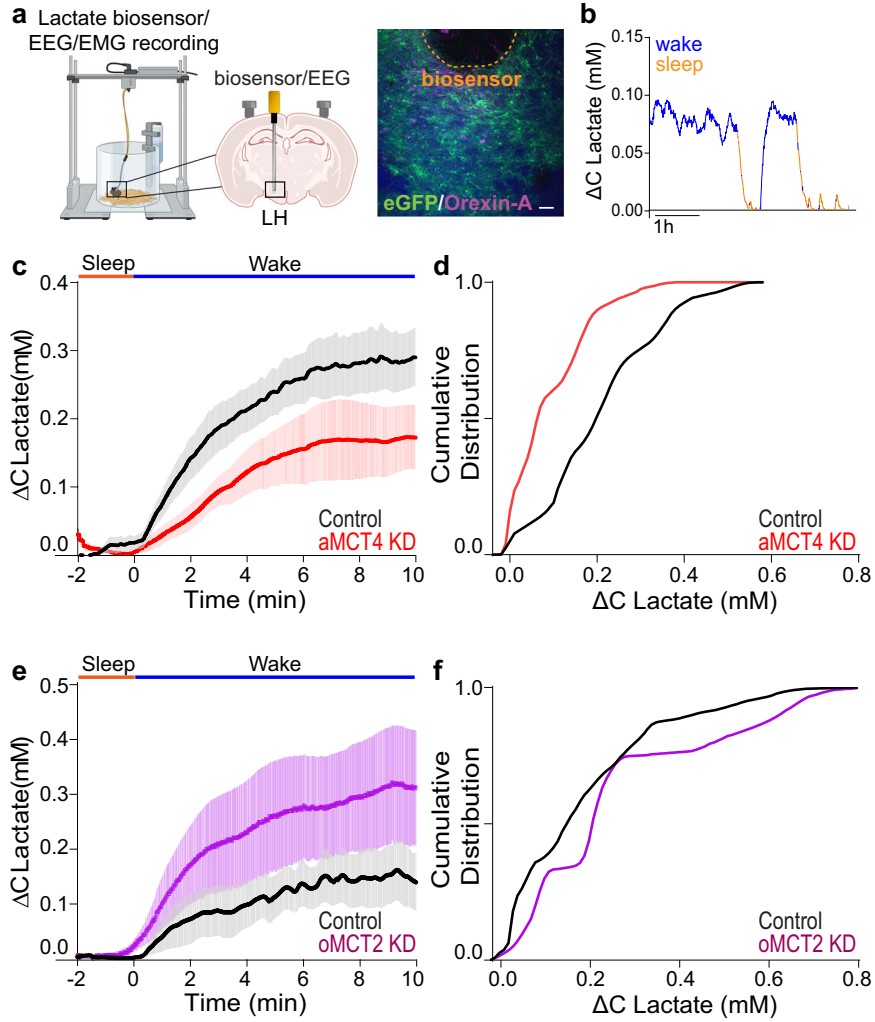

**Fig. 5 | Deletion of astrocytic MCT4 reduces extracellular lactate surges during wakefulness, while deletion of orexinergic neuron MCT2 increases extracellular lactate surges during wakefulness. a** Schematic representation of lactate biosensor recording/placement coupled with EEG/EMG, and immunofluorescence micrograph showing positioning of lactate biosensor (highlighted by orange line) in the LH near Orexin-A positive cells (magenta) and eGFP positive astrocytes in green. Scale bar = 100 μm. **b** Representative trace of extracellular lactate recording during the dark phase color coded according to EEG analysis. Mean stereotypical traces for control and aMCT4 KD mice (**c**), and oMCT2 KD mice (**e**) upon wakefulness (blue line "Wake"). Pooled data are shown as mean ± SEM. EEG=electroencephalogram; EMG=electromyography (**d**, **f**) cumulative frequency plots (Two-sample Kolmogorov–Smirnov test) of lactate concentrations during all wakefulness events in the first hours of the dark phase (ZT12-18). aMCT4 KD mice (red, $n = 4$) compared to controls mice (black, $n = 5$) (Kolmogorov–Smirnov test, $p = 3.4135 \times 10^{-57}$); (**f**) oMCT2 KD (violet, $n = 6$) mice compared to controls mice (black, $n = 5$) (Kolmogorov–Smirnov test, $p = 1.887 \times 10^{-17}$). Source data are provided as a Source Data file.

pharmacological inhibition of glycogenolysis, by perfusing DAB into the LH, reduced prolonged wakefulness and increased NREM sleep during the dark phase, when mice are mostly awake and active. Consistent with our hypothesis, delivery of lactate into the LH of DAB-treated mice fully restored wakefulness stability. This could be especially relevant because norepinephrine, a known enhancer of arousal, stimulates glycogen metabolism and lactate release from astrocytes[26,37]. Whether astrocytes are the target of noradrenergic signaling into the LH needs to be fully elucidated. We then demonstrated that pharmacological inhibition of MCTs by perfusing 4-CIN into the LH, decreased wakefulness during the dark phase. This finding confirms the role of lactate exchange in the regulation of wakefulness during this phase. We also detected an effect during the light phase, and it is noteworthy that we cannot exclude off-target effects of 4-CIN, including extracellular and intracellular acidification due to impaired proton transport inhibition[38] of mitochondrial pyruvate transport[39] or effects on other cell types, rather than astrocytes expressing MCTs.

To further investigate and dissect the cell-specific role of MCTs we employed transgenic mouse lines to selectively downregulate astrocytic MCTs. Downregulation of astrocytic MCT4 resulted in reduced consolidated wakefulness during the dark phase, rescued by delivery of exogenous lactate. Such results are in line with a previous hypothesis suggesting a role of MCT4 as an activity-dependent lactate exporter[40] and with recent evidence showing MCT4 as the only isoform able to export lactate extracellularly despite elevated levels[41]. Accordingly, lactate concentration in various brain regions is higher during the dark phase compared to the light phase[42–44]. To further investigate the impact of astrocytic MCT4 downregulation on extracellular lactate changes we employed lactate biosensors measurements coupled with EEG/EMG recordings. In line with previous studies[16,42,45], we observed, daily lactate fluctuations, with extracellular lactate reaching a low basal level during sleep and consistently increasing during wakefulness, suggesting that this phenomenon might be common across different brain regions. While our in vivo lactate measurements did not allow us to measure absolute

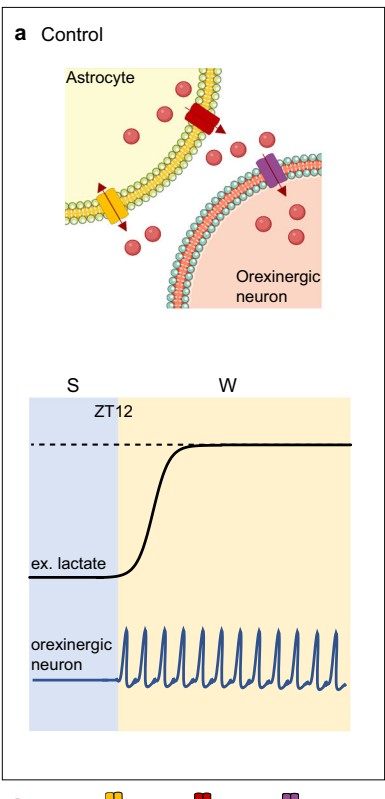
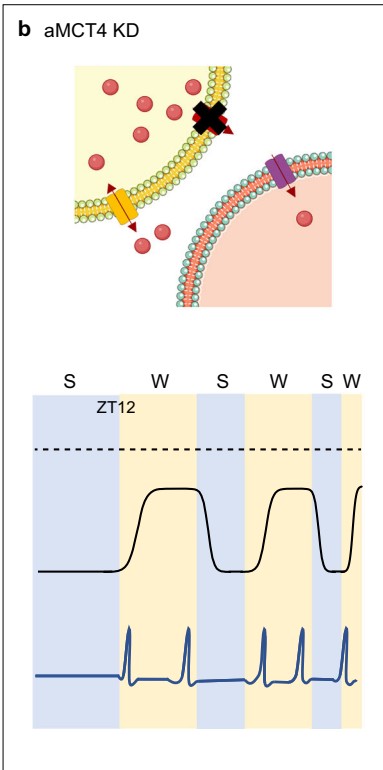
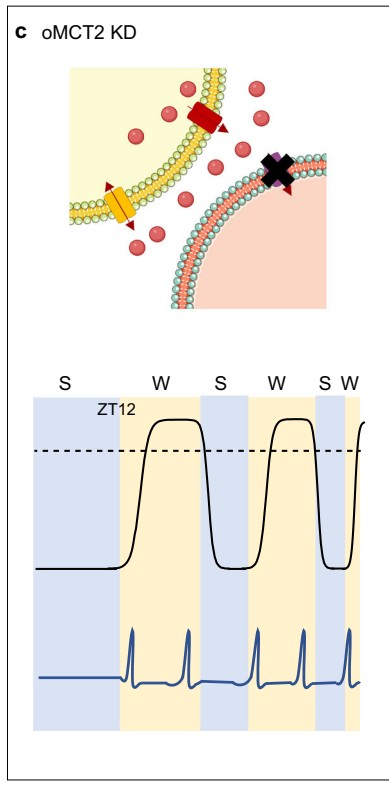

● Lactate ▮ MCT1 ▮ MCT4 ▮ MCT2

**Fig. 6 | Astrocytic-derived lactate supports orexinergic activity and consolidates wakefulness. a** During the transition from sleep to wake lactate is released through astrocytic MCT4, which serves as an energy source and modifies orexinergic excitability as well as activity, thus promoting wakefulness consolidation. **b** Deletion of MCT4 results in a reduction in lactate release during wakefulness, leading to a decrease in orexinergic tonic firing and subsequent fragmentation of wakefulness. **c** Deletion of MCT2 hinders the import of lactate into orexinergic neurons, resulting in decreased orexinergic activity and compromised stability of wakefulness. ZT= Zeitgeber Time; S= sleep; W= wake.

concentrations, MCT4 downregulation resulted in reduced surges of extracellular lactate upon wakefulness, corroborating its role as an exporter during the highly active dark phase.

The ANLS hypothesis has long proposed that astrocytes are capable of exporting lactate into the extracellular environment as a metabolic substrate to sustain neuronal activity during high energy-demanding tasks[10,11,15,24,26,46,47]. Does the reduction of extracellular lactate in aMCT4 KD mice have a downstream effect on neuronal excitability? We hypothesized that arousal instability observed in aMCT4 KD mice was associated with a decreased activity of orexinergic neurons due to impairment in lactate shuttling. Extending previous findings, we found that downregulation of astrocytic MCT4 dramatically hyperpolarized and decreased the firing rate as well as $R_{in}$ of orexinergic neurons. First, the decreased orexinergic excitability is in line with the wakefulness instability observed in vivo as orexinergic neurons play a major role in maintaining and promoting wakefulness[30], increasing their activity during active wakefulness[18,19], and showing frequency-dependent control of vigilant states. Second, these data confirm that lactate released through MCT4 is a necessary metabolic substrate to sustain orexinergic excitability even in the presence of physiological glucose concentration. Consistent with this idea, we found that perfusion of extracellular lactate was sufficient to rescue their activity in brain slices from aMCT4 KD mice. In contrast with recent experimental findings and modeling interpretations[48], in our experimental conditions, we exclude that the effect of lactate is confounded by changes in osmolarity or by intracellular acidification due to impaired proton transport[38,49] since we kept all the solutions isosmotic, and we buffered intracellular pH changes with HEPES.

Astrocytic MCT1 has been implicated in basal lactate release[40], but its exact role in brain functions has yet to be elucidated. In our study, we observed that MCT1 downregulation led to increased fragmented wakefulness during the light phase accompanied by increased events during the dark phase. Such results would be in line with the characterization of MCT1 as an essential exporter in low ambient lactate levels as described during the resting phase[44], and not as efficient as MCT4 at higher extracellular concentrations during the active phase[41].

Neurons across the brain mainly express MCT2[8], and have been shown to predominantly import lactate[11,26]. By selectively downregulating MCT2 in orexinergic neurons, we found that mice were unable to maintain prolonged wakefulness in vivo during the dark phase. In vitro, deletion of MCT2 in orexinergic neurons decreased firing rate and $R_{in}$ while hyperpolarizing the membrane potential of these neurons. By contrast to what was observed in astrocytic MCT4 downregulation, extracellular lactate was not able to change neuronal activity neither in MCT2 KD mice nor in the presence of 4-CIN, highlighting the crucial role of importing lactate. Further supporting the role of orexinergic neurons as lactate sink, extracellular lactate measurements showed increased surges upon wakefulness in MCT2 KD mice, likely the result of lactate accumulation in the extracellular milieu.

In line with previous evidence demonstrating fragmented wakefulness during the dark phase in orexin-deficient mouse models[50,51], we did not observe any phenotype in the light phase in aMCT4 KD and oMCT2 KD mice. This finding confirms the role of orexin neurons and suggests that other neurons in the same area, such as melanin-concentrating hormone neurons (MCH), may not be dependent on astrocytic lactate. In addition, we also identified a consistent role for

lactate as a fuel during the first 6 h of the dark phase. The lack of effects in the latter part (ZT18-24) may indicate reduced orexinergic activity, potentially sustained by glucose or other metabolites. Notably, in our experiments, orexinergic neurons can still use glucose as an energetic source, emphasizing that glucose and lactate are not interchangeable in modulating orexinergic activity and wakefulness. Although embryonic manipulation of MCT2 in orexinergic neurons could be considered a confounding strategy, the re-expression of MCT2 during the adult stage resulted in the recovery of the wakefulness phenotype. This observation suggests the absence of any developmental impairments.

Finally, orexin neurons express functional $K_{ATP}$ channels[20,21,33,52] known to modulate their activity. As expected, we found that tolbutamide, a selective $K_{ATP}$ blocker, rescued orexinergic firing in both astrocytic and orexinergic neuron MCTs knockdown mice, indicating $K_{ATP}$ channel as the downstream target of the lactate effect. These data are consistent with similar observations made in cortical neurons showing that lactate uptake (via MCT2) and its utilization as energy substrate is key to maintaining their excitability via regulation of $K_{ATP}$ channels[32].

Collectively, data obtained from MCT2 knockdown mice reinforce the hypothesis that lactate supports orexinergic firing due to its internalization through MCT2 and its use as an energy substrate, ruling also out the possibility that lactate could exert its effect through the binding to an extracellular receptor.

This is of particular interest since alterations of the sleep-wake cycle and orexinergic activity, as well as degeneration of these neurons, have been reported in several degenerative diseases[53] and aging[54]. For instance, one of the earliest symptoms in preclinical Alzheimer's disease (AD) is excessive daytime sleepiness and difficulty in maintaining long periods of consolidated wakefulness[55,56] resembling the fragmented waking pattern we identified in our transgenic mice lines after deleting astrocytic MCT4 and orexinergic MCT2 in the LH. Concurrently, brain metabolic dysfunctions, marked by abnormal glucose metabolism and glycolytic rate, as well as altered expression of glucose transporters and MCTs, have been also reported in both patients and murine models of AD[57–61]. However, whether these metabolic alterations correlate with the sleep disturbances observed in these pathologies remains to be elucidated. Thus, our findings have the potential to shed light on the mechanisms underlying sleep disturbances in pathological conditions, paving the way for translation studies aimed at modulating MCTs and brain lactate concentration.

In conclusion, our results provide evidence for the importance of the astrocytic lactate shuttle in regulating sustained wakefulness. By using cell-specific downregulation of MCTs, we demonstrate that astrocytes are a major contributor to lactate export into the extracellular environment where it is available for uptake by orexinergic neurons, modulating their activity and ultimately regulating wakefulness.

## Methods

### Animals, housing, genotyping

All animal experiments were conducted in accordance with the guidelines of the Animal Care and Use Committee of Tufts University. Protocol number: B2022-120.

Male and female mice were bred and housed on a 12/12 light/dark cycle, given standard chow and water ad libitum. Zeitgeber time (ZT) scale, that sets the origin of the 24 h period (ZT0) to the onset of the light-phase, has been used to allow comparison among studies independently of the actual clock-time settings of animal facilities. Ambient temperature was maintained at $23 \pm 1\,°C$, and humidity was kept within the range of 40% to 60%.

Homozygous MCT4[floxed/floxed], MCT2[floxed/floxed] and MCT1[floxed/floxed] mice originated from Dr. L. Pellerin's lab (Department of Physiology, University of Lausanne, Lausanne, Switzerland). Upon receival, mice

were cleared from quarantine, backcrossed with C57BL/6 mice, and bred in our facility.

MCT4[floxed/floxed] mice present exons 3,4 and 5 of the Slc16a3 gene flanked by LoxP sites. The floxed allele showed a 516 bp band, while the endogenous Slc16a3 allele showed a 388 bp band (Supplementary Fig. 2a). Allele from KD mice showed a 368 bp band, while intact floxed allele showed a 446 bp band (Supplementary Fig. 2a).

MCT1[floxed/floxed] mice present exon 5 of the Slc16a1 gene flanked by LoxP sites. The floxed allele showed a 420-base pair (bp) band, while the endogenous Slc16a1 allele showed a 280 bp band, while allele from KD mice showed a 437 bp band, while intact floxed allele showed a 227 bp band (Supplementary Fig. 2f).

MCT2[floxed/floxed] mice present exons 4 and 5 of the Slc16a7 gene flanked by LoxP sites. The floxed allele showed a 436 bp band, while the endogenous Slc16a7 allele showed a 308 bp band, while allele from KD mice showed a 227 bp band, while intact floxed allele showed a 437 bp band (Supplementary Fig. 5a).

Orexin-IRES-Cre mice were generated, validated, and kindly provided by Dr. Dong Kong and crossed with MCT2[floxed/floxed] mice.

To test the specificity of our viral promoter GFAP (0.7), Ai14 mice (#007909, Jackson Laboratory) have been kindly donated from Dr. Yongjie Yang.

### Adeno-virus vector stereotaxic injection

Eight weeks old male and female MCT1[f/f], MCT4[f/f] and Orexin-IRES-Cre x MCT2[f/f] (oMCT2 KD) mice were anesthetized with isoflurane, injected with buprenorphine (0.5 mg/kg) and placed into a stereotaxic frame. Two small holes were bilaterally drilled in the skull (LH coordinates: AP = −1.6, ML = ± 0.9 mm). A 2 μL Neuro 7002 syringe (Hamilton) was filled with AAV ($1 \times 10^9$ virus genomes per μL) and the needle lowered into the holes to reach the LH (LH coordinates: DV = − 5.35 mm). 1 μL of vector per hemisphere was injected at a speed of 100 nL/min using a Microsyringe Pump Controller Micro4 (Harvard Apparatus). MCT4[f/f] or MCT1[f/f] mice received either AAV-PhP.eB-GFAP (0.7)-EGFP-T2A-iCre virus (aMCT4 KD or aMCT1 KD mice) or AAV-PhP.eB-GFAP (0.7)-EGFP virus (control mice) (#VB1131 Vector Biolabs) or vehicle only (control mice) (ACSF, #3525 Tocris). oMCT2 KD received AAV-PhP.eB-CAG2-DIO-mSlc16a7-P2A-GFP virus (AAV-272101, Vector Biolabs) to re-express MCT2 in these mice. To visualize orexin-expressing neurons for electrophysiological recordings oMCT2 KD and control mice received AAV9-CAG-DIO-mCherry virus (VB1326, Vector Biolabs). 7 weeks old female and male Ai14 mice have been bilaterally injected with AAV-PhP.eB-GFAP(0.7)-EGFP-T2A-iCre to test for viral transduction specificity.

### EEG/EMG and lactate biosensor surgery, recording and analysis

4 or 10 weeks after viral injections MCT1[f/f] or MCT4[f/f] mice, depending on the experimental setting, and 7 weeks old oMCT2 KD and related controls mice were anesthetized with isoflurane, injected with buprenorphine (0.5 mg/kg) and placed into a stereotaxic frame. EEG/EMG implantation surgeries have been performed as previously described[21]. Briefly, prefabricated EEG/EMG headmounts (8201, Pinnacle Technology, Lawrence, KS) were secured to the skull with four stainless steel EEG screws (8209, Pinnacle Technology, Lawrence, KS) into the pilot holes. Silver epoxy was applied to ensure electrical connectivity between the electrodes and the headmount. EMG leads were inserted bilaterally into the nucal muscles. The headmounts were then secured to the skull with dental acrylic. 10 days after surgery, mice were moved into an insulated sound-proof chamber and placed into individual Plexiglas circle boxes (Pinnacle Technology, Lawrence, KS) containing water and food ad libitum and headmounts were plugged to a light-weight EEG preamplifier (Pinnacle Technology) that allowed freely moving. Mice were left 3 days for habituation prior to data collection and maintained on a 12:12 light/dark cycle. EEG signals were acquired at a frequency sampling with 400 Hz using Sirenia software (Pinnacle

Technology, Lawrence, KS). Sleep stages were scored visually at 4 s epochs by a trained experimenter using SleepSign for Animal software (Kissei Comtec). Wakefulness (W) was defined as low-amplitude, high frequency EEG and high EMG activity; REM sleep was characterized by low amplitude, desynchronized EEG with low EMG activity; and non-rapid eye movement (NREM) sleep consisted of high-amplitude, low frequency EEG with little EMG modulation. The dark phase was divided into ZT12-18 and ZT18-24 based on prior observations[21,62,63] and studies showing differential activity in the dark phase[64–66].

For in vivo lactate biosensor experiments, 10 weeks after viral injections MCT4[f/f] mice and 7 weeks old oMCT2 KD and related control mice were anesthetized with isoflurane, injected with buprenorphine (0.5 mg/kg) and placed into a stereotaxic frame. Three small holes were drilled, one into the frontal area and two into the parietal area, and EEG screws with wire leads (8403, Pinnacle Technology, Lawrence, KS) were inserted and manually rotated into the pilot holes. One additional hole was drilled in the skull (LH coordinates: AP = −1.6, ML = 0.9 mm) and the guide cannula was lowered into the hole to reach the LH upon insertion of the biosensor (LH coordinate DV = −3.35 mm). The cannula and screws were secured to the skull with dental acrylic. EEG screw wire leads were subsequently soldered to prefabricated EEG/EMG/Bio headmounts (8402, Pinnacle Technology, Lawrence, KS); EMG leads were inserted bilaterally into the nucal muscles, and headmounts were also secured with dental acrylic. 10 days after surgery, mice were moved into an insulated sound-proof chamber and let habituated as described above. 7–8 h before the dark phase (ZT12) lactate biosensors were in vitro precalibrated with known concentration of L-lactate and the interferent Ascorbic Acid, as per manufacturer's instruction, and inserted into the guide cannula. EEG and EMG signals were acquired as described above. Lactate data were also acquired through Sirenia software. EEG/EMG/biosensor recordings continued for ~18 h, followed by in vitro biosensor postcalibration with known concentration of L-lactate and the interferent Ascorbic Acid, as per manufacturer's instruction. Biosensors that were sensitive to the interferent Ascorbic Acid during pre or postcalibration were excluded from the analysis.

For lactate measurements analysis: The current trace (nA) recorded using the Pinnacle's lactate biosensor was corrected as described in the Supplementary Materials section. The resulting trace represented the lactate concentration changes (mM) compared to the baseline (set at 1 mM). The sleep (S) and wake (W) events were identified during a 6-h period (from ZT12 to ZT18) using the EEG/EMG signals, as previously described, and used to define the corresponding values of lactate's change in each state. For a given state (either S or W) and for each genotype, the values of lactate's change were combined and plotted as histograms (bin width: 0.01 mM). To compare the CTRL vs KD conditions, the values for each group for a given state were subsequently used to compute the Kernel probability distribution (pd) using the fitdist.mat function in MATLAB; the resulting Kernel pd was used to reconstruct a smaller dataset (considering the sensitivity of the Kolmogorov−Smirnov test to the sample size of the analyzed vectors; $n = 400$) using the random.mat function in MATLAB and the pd as an input. The reduced datasets were compared with a two-sample Kolmogorov−Smirnov test (kstest2.mat function in MATLAB) testing one of the two alternative hypotheses (using the 'Tail' option in the kstest2.mat function): (1) H1: values in CTRL distribution greater than values in the KD distribution or (2) H1: values in CTRL distribution smaller than values in the KD distribution. H0: values in CTRL KD belongs to the same distribution. Stereotypical traces were visually identified as traces reaching stable values of lactate's change during the W state for 10 min and preceded by 2 min of stable lactate's change (during S state). The traces were then aligned, with 0 min as the S-W transition and averaged for each genotype. The full trace correction description can be found here: (https://osf.io/fwru5/?view_only=48ee97f56e424b219f66ef3d9d7ed20d).

## Drug injections in freely behaving mice

10 weeks old MCT4[f/f] mice or 7 weeks old C57BL/6 were anesthetized with isoflurane, injected with buprenorphine (0.5 mg/kg) and placed into a stereotaxic frame. Surgeries were performed as previously described. Briefly, two small holes were bilaterally drilled in the skull (LH coordinates: AP = −1.6, ML = ± 0.9 mm). Bilateral brain cannula was stereotaxically implanted into the LH (Guide cannula #8IC235G18XXC with 4 mm pedestal length, internal cannula #8IC235ISPCXC 1.25 mm length below the pedestal, P1 Technologies). Mice were then implanted with EEG/EMG electrodes, as described earlier. After 10 days of post-operative recovery, mice were placed into individual Plexiglas circle boxes and their micro-connectors plugged to EEG preamplifier, as described above. The following drugs have been used: sodium L-lactate (5 mM, equicaloric to 2.5 mM glucose, Sigma, #L6022),1,4-Dideoxy-1,4-imino-D-arabinitol hydrochloride (DAB, 10 mM, Sigma, #D1542), or Alpha-cyano-4-hydroxycinnamic acid (4-CIN, 1,5 mM, Sigma, #C2020).To allow for continuous and controlled drug delivery, osmotic mini-pumps (model 1002; Alzet; flow rate, 0.25 µL/h; 2-wk duration) were secured with the flow moderator, and primed overnight in 0.9% saline solution at 37 °C before to be connected to the brain cannula by flexible catheter tubing (#51158, Stoelting). Osmotic mini-pumps were externalized as previously described[21]. Briefly, cannulas were placed in a sealed Eppendorf filled with 0.9% saline solution and maintained at 37 °C with an iBlock Mini Dry Bath (Midsci) placed on top of the cage. The catheter tubing was then filled with aCSF (Harvard Apparatus, 59-7316), L-lactate, DAB or 4-CIN, depending on the experimental setting, and a small air bubble was inserted into the catheter tubing during its connection with the pump to monitor the flow rate. The flexible catheter tubing was carefully aligned and attached to the EEG cable. Mice were left 3 days for habituation prior to drug delivery, recordings and data collection, and maintained on a 12:12 light/dark cycle. Each drug was delivered for 4 consecutive days, EEG/EMG recordings were acquired between day 3 and 4 of drug delivery and data analyzed as described above.

## Brain slice preparation

Brain slice experiments were performed on male and female adult mice (2−6 months depending on the experimental setting). Briefly, mice were anaesthetized with isoflurane, and after decapitation, the brain was rapidly removed and put in ice-cold (−2° to −4 °C) carbogen-bubbled (O2 95%/CO$_2$ 5%) artificial cerebrospinal fluid (ACSF) containing the following (in mM): 120 NaCl, 3.2 KCl, 1 NaH2PO4, 26 NaHCO3, 1 MgCl2, 2 CaCl2, 2.5 glucose (osmolarity adjusted to 300 mOsm with sucrose, pH 7.4). After removal of the cerebellum, the brain was glued and coronal hypothalamic slices (350 µm) containing the LH were cut using a vibratome (VT1200S; Leica). Before recording, slices were incubated at 33 °C for a recovery period of 45 min. After recovery, slices were placed in a submerged recording chamber (Warner Instruments) and continuously perfused (2 mL/min) with oxygenated ACSF. The glucose concentration used for recording was 2.5 mM unless otherwise noted. Slices were used for maximally 4 h after dissection. Experiments were performed at room temperature 21° to 24 °C.

## Patch-clamp recordings and cell phenotyping

Orexin neurons were visualized with an 16x objective and 2x magnification in an upright Nikon Eclipse FN1 microscope, equipped with an arc-discharge mercury lamp (Nikon intensilight C-HGFI), by using infrared differential interference contrast (IR-DIC). For patch-clamp recordings, oMCT2 KO and control mice were used at 8 weeks old (2 months). aMCT4KO and control mice were used 12 weeks post viral injection (5 months old).

In oMCT2 KD and control slices, orexin neurons were selected based on mCherry expression.

In aMCT4 KD and control slices, viral transduction was confirmed based on GFP expression brain slices showing no infection were discarded. Orexin neurons were discriminated from melanin-concentrating hormone neurons (MCH) residing in the same region for their electrophysiological fingerprints[21].

Whole-cell patch-clamp recordings were performed in current-clamp mode by using a Multiclamp 700B amplifier (Molecular Devices). Data were filtered at 1 kHz and sampled at 5 kHz with Digidata 1322 A interface and Clampex 9.2 and 10.6 from pClamp software (Molecular Devices).

Pipettes (from borosilicate capillaries; World Precision Instruments) had resistance of 6−7 MΩ when filled with an internal solution containing the following (in mM): 123 K-gluconate, 2 $MgCl_2$, 8 KCl, 0.2 EGTA, 4 Na2-ATP, 0.3 Na-GTP, and 10 HEPES, pH 7.3 with KOH.

Cell-attached patch-clamp recordings were performed in neurons in voltage-clamp mode to record the spontaneous firing activity avoiding dilution of the intracellular compartments. All recordings were analyzed with Clampfit 9.2 and 10.6 from pClamp software (Molecular Devices). Mean firing rate were calculated from 2 min of stable recordings.

All drugs were applied to the perfusing system (bath application) to obtain the final concentrations indicated unless otherwise stated. Alpha-cyano-4-hydroxycinnamic acid (4-CIN), sodium L-lactate and tolbutamide were obtained from Sigma.

Neurobiotin Tracer (1 mg/mL; #SP-1120, Vector Laboratories) was also added to the internal solution during whole-cell recordings for post hoc immunohistochemical phenotyping. Briefly, slices were fixed overnight with 4% paraformaldehyde (PFA), then rinsed several times with a PBS solution and permeabilized and immunoblocked with 0.5% Triton X-100 and 5% NGS (Normal Goat Serum) in PBS. For immunostaining of orexin neurons, slices were incubated with rabbit anti-Orexin-A (1:500; #AB6214, Abcam) for three days at 4 °C. The primary antibody was visualized with anti-rabbit Alexa Fluor 633 (1:500; #A21071, Invitrogen). Sections were also incubated with streptavidin 546 (1:500; #S11225, Invitrogen) to label neurons infused with Neurobiotin, then mounted with Vectashield antifade mounting medium with DAPI (Vector Laboratories) and visualized using a confocal laser scanning microscope (Nikon A1) with a 40x oil immersion lens (NA1.0).

## Western Blot

Due to the confined size of the lateral hypothalamus, we quantified protein downregulation from the entire isolated area, thus including all cell types within the tissue. To reduce the number of mice used, proteins were collected from 30 µm sections from perfused mice (see section below). LH were isolated after confirmation of viral expression based on GFP visualization. Total protein was then extracted by resuspending the isolated tissue in radioimmunoprecipitation assay (RIPA) buffer (10 mM Tris HCl, 0.1 M, pH 7.2; 1% sodium deoxycholate; 1% Triton X-100; 3% sodium dodecyl sulfate [SDS]; 150 mM NaCl, 1.5 M; 1 mM EDTA, pH 8.0, 0.5 M (#AM9260G, Thermo Fisher Scientific); 1 mM phenylmethanesulfonyl fluoride (#93482, Sigma); Complete Protease Inhibitor cocktail (#539131, Millipore Roche); Halt Phosphatase Inhibitor (#1862495, Thermo Fisher Scientific)) and stored at −20 °C until usage. Samples were then heated for 20 min at 100 °C and then let sit on ice for additionally 20 min prior to sonication. Samples were then subjected to sonication (Soniprep 150, MSE) and spun at 21,000 g for 10 min, supernatants were then collected for protein quantification. Protein quantification was performed using a Pierce BCA Protein Assay kit (#23227, Thermo Scientific). 10 µg of total protein were mixed with NuPAGE LDS Sample Buffer (#NP007, Life Technologies), NuPAGE Sample Reducing Agent (#NP0009, Life Technologies), and distilled water prior to being heated at 95 °C for 5 min. Proteins were separated by SDS-PAGE in a 4%−12% Bis-Tris gel (#NP0336, Thermofisher) using a Novex Bolt Mini Gel system and NuPAGE™ MOPS SDS Running Buffer (NP0001, Thermofisher) before being transferred onto Immobilon-P polyvinylidene fluoride membranes (#IPVH00010, 0.45 mm pore size; Millipore) in NuPAGE™ Transfer Buffer with methanol 20% (NP0006, Thermofisher) for 1 h 40 min at 4 °C using a Novex Bolt Mini Blot Module. SeeBlue Plus2 standard (#LC5925, Life Technologies) was used to estimate protein sizes, and transfer was confirmed by Ponceau S (#BP103-10, Fisher Biotech) staining. Immunoblot was obtained by first blocking membranes for 1 h at room temperature with a solution of 0.1% Tween 20 and 5% Bovine Serum Albumin (BSA) (#5217, Tocris) in 1× PBS (pH 7.4) and then incubated with primary antibodies overnight at 4 °C: rabbit anti-MCT1 (1:10.000, #20139, Proteintech) and rabbit anti-MCT4 (1:1000, #NBP1-81251, Novus Biologicals). After washing with 1× PBS containing 0.1% Tween 20, the membranes were incubated with the species-appropriate horseradish peroxidase-conjugated secondary antibodies for 1 h. at room temperature at a 1:15,000 dilution in blocking solution. Immunoreactivity was revealed using SuperSignal™ West Pico PLUS Chemiluminescent Substrate (#34577, Thermofisher) and imaged using a Fujifilm LAS 4000 Gel Imager system with ImageQuant LAS 4000 software (Fujifilm). If needed, antibodies were stripped from membranes prior to incubation with another primary antibody or prior to incubation with mouse anti-β-actin (1:1000, Sigma, #A1978) in a stripping solution with 0.7% β-mercaptoehtnaol (#63689, Sigma). Densitometry measurements were performed using Fiji, with each protein band being normalized to their respective β-actin. Full membrane images can be found in the following link: https://osf.io/fwru5/?view_only=48ee97f56e424b219f66ef3d9d7ed20d.

## Tissue processing and immunohistochemistry

Mice were anesthetized with isoflurane and transcardially perfused with saline (sodium chloride 0.9%), followed by cold 4% PFA in PBS pH 7.4. The brains were post-fixed in the same solution for 12 h at 4 °C and then cryoprotected for at least 24 h in 30% sucrose (Sigma) in 1× PBS at 4 °C. Coronal sections containing the LH were cut serially on a freezing microtome (SM2000R; Leica) at a thickness of 30 µm and then stored in an antifreeze solution at −20 °C until immunostaining.

For evaluation of cell-specificity upon AAV injection: 4 serial sections 120 µm apart were used and Cre-induced Td-Tomato expression was evaluated in Ai14 mice. Free-floating sections were washed twice in 1x PBS for 10 min and then incubated with 10% normal goat serum (NGS, #5425, Cell Signaling) blocking solution (1x PBS, 0.3% Triton) for 1 h. at room temperature, followed by incubation with primary antibodies (chicken anti-GFAP 1:500, Abcam, #AB4674; or rabbit anti-Orexin-A 1:500, Millipore, #AB3704) overnight at 4 °C on a shaking platform. Sections were rinsed three times with 1x PBS solution and then incubated with secondary antibodies (633 goat anti-rabbit, Invitrogen, #A21071; or 633 goat anti-chicken, Ivitrogen, #A21103) for 2 h. at room temperature. All sections were counterstained with DAPI, mounted on slides with Vectashield antifade mounting medium (#H-1800, Vector Laboratories). Fluorescent images were acquired with a confocal laser scanning microscope (Nikon A1) using a 20x objective. Image analysis and counting of cells expressing Td-tomato with Orexin-A or with GFAP were performed within the LH by using Cell-Counter plug-in in Fiji software. Data are expressed as percentage of double positive cells over the total number of Td-Tomato positive cells. More than 50 cells per animal were counted.

For cannula placement verification: 5 serial sections 120 µm apart were used for assessing cannula placement (as well as biosensor placement). Free-floating sections were washed twice in 1x PBS for 10 min and then incubated with 10% normal goat serum (NGS) blocking solution (1x PBS, 0.3% Triton) for 1 h at room temperature, followed by incubation with primary antibodies (chicken anti-GFP 1:500, Abcam, AB13970; or rabbit anti-Orexin-A 1:500, Abcam, #AB6214) overnight at 4 °C on a shaking platform. Sections were rinsed three times with 1x PBS solution and then incubated with secondary antibodies (546 goat anti-rabbit, Invitrogen, #A11035; and/or 488 anti-chicken, Invitrogen,

#A11039) for 2 h. at room temperature. All sections were counter-stained with DAPI, mounted on slides with Vectashield antifade mounting medium and checked at an epifluorescence microscope Nikon E800. Bilateral cannula placement representative images were acquired using an epifluorescence microscope (Keyence BZ-X700) with a 10x objective and stitched with Keyence software. Biosensor placement representative images were acquired with a confocal laser scanning microscope (Nikon A1) using a 20x objective. For beau-tification purpose, brightness of representative images has been changed to increase contrast using Fiji software. (https://osf.io/fwru5/?view_only=48ee97f56e424b219f66ef3d9d7ed20d).

To examine cell-targeting specificity, Cre-induced mCherry expression in orexin neurons, was evaluated in 6 brain sections from 4 Orexin-cre mice injected in the LH with the AAV9-CAG-DIO-mCherry virus.

Free-floating sections were washed twice in 1x PBS for 10 min and then incubated with 10% normal goat serum (NGS, #5425, Cell Sig-naling) blocking solution (1x PBS, 0.3% Triton) for 1 h at room tem-perature, followed by incubation with primary antibodies (mouse anti-Orexin-A 1:200, R&D Systems, #MAB763, and rabbit anti-mCherry 1:500, Thermo Fisher, #600-401-P16) overnight at 4 °C on a shaking platform. Sections were rinsed three times with 1x PBS solution and then incubated with secondary antibodies (488 goat anti-mouse, Invitrogen, #A-11001; and 546 goat anti-rabbit, Invitrogen, #A11035) for 2 h at room temperature. All sections were counterstained with DAPI, mounted on slides with Vectashield antifade mounting medium. Confocal stacks were acquired using a confocal laser scanning micro-scope (Nikon A1) using a 40x oil immersion lens (NA1.0). Cell counting was performed using Fiji by scanning through the z-planes manually, DAPI staining was used to colocalize each cell's nucleus with an orexin-specific marker (Orexin-A) and the reporter (mCherry). In the results section shown in Supplementary Fig. 7, we found that $94\% \pm 2.1\%$ of mCherry expressing cells (107 cells counted, from 6 brain sections) were Orexin-A + . On average $85.6\% \pm 4.9\%$ of all orexin neurons (Orexin-A + , 120 cells from 6 brain sections) were mCherry+ (express-ing Cre recombinase). Data are expressed as mean ± SEM of brain sections analyzed.

For orexin neuron quantification: 5 serial sections 120 μm apart from Orexin-Cre negative or Orexin-Cre positive mice were used for quantification of orexin neurons. Sections were pre-treated with per-oxidase 3% for 15 min and then incubated for 1 h at room temperature in the blocking solution (1× PBS, 10% NGS, 0.3% Triton X-100), followed by incubation with Rabbit anti-Orexin-A primary antibody (rabbit anti-Orexin-A 1:500, Abcam, #AB6214) in blocking solution and incubated at 4 °C overnight. The following day, after washing the sections with 1× PBS, sections were incubated with biotinylated secondary antibody (biotinylated goat anti-rabbit, 1:500, Abcam, #AB6720) for 1 h at room temperature. Sections were then incubated with A and B components of Vectastain Elite ABC kit (PK-4000, Vector Laboratories) for 1 h at room temperature, and finally the reaction was developed using 3,3′-diaminobenzidine (DAB) following the supplier's instructions (SK4100 Vector Laboratories). The reaction was blocked by washing the sec-tions with distilled water and mounted with Poly-Mount (#08381, Polysciences). Brightfield images were acquired with Nikon E800 microscope, and of Orexin-positive cells within the LH were counted by using Cell-Counter plug-in in Fiji software. Data are expressed as average of total cells per animal.

### Statistical analysis
For slice electrophysiology, normal distribution was assessed using Kolmogorov–Smirnov test. Comparisons between two groups were conducted with the paired or unpaired Student's $t$ test or the non-parametric Mann–Whitney test as appropriate. Repeated-measures ANOVAs followed by Tukey's test were used when multiple measure-ments were made over time in the same groups, the data were analyzed by fitting a linear mixed-effects model in case of missing values. All values and related statics can be found in Source Data file.

EEG/EMG: Repeated-measures ANOVA was used when multiple measurements were made over time in the same groups followed by Tukey's post hoc multiple comparisons tests. Comparisons between two groups were conducted with the paired or unpaired Student's $t$ test. Lactate biosensor traces were analyzed by using two-sample Kol-mogorov–Smirnov test. The level of significance was set at $p < 0.05$. Statistical analysis was carried out using the Prism7 and Prism8 software (GraphPad Software, Inc). All values, sample size and related statistics for 24EEG analysis can be found in the Statistics Table (https://osf.io/fwru5/?view_only=48ee97f56e424b219f66ef3d9d7ed20d).

Schematic representations throughout the manuscript have been created with licensed BioRender.com.

### Reporting summary
Further information on research design is available in the Nature Portfolio Reporting Summary linked to this article.

## Data availability
All data generated or analyzed during this study are included in this article and its supplemental materials. Source data are provided as a Source Data file. Source data are provided with this paper.

## Code availability
The code used for analyzing lactate biosensor traces is publicly avail-able at the following repository: https://osf.io/fwru5/?view_only=48ee97f56e424b219f66ef3d9d7ed20d.

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

## Acknowledgements

We are grateful to Angela Capriglione for the management of mice colonies and technical support. We are also grateful to the University of Lausanne for the agreement to export and use the MCT1, 2 and 4 floxed mice. P.G.H. was supported by NIH grant (R01 NS107315-04). L.P. was supported by Swiss FNS/French ANR grant (FNS 310030E-164271/ANR-15-CE37-0012) and ANR grant (ANR-21-CE44-0023-01). The authors declare no competing interests.

## Author contributions

P.G.H. conceived and supervised research. A.B and M.C equally contributed to this work. The order of these authors reflects alphabetical order. A.B. and M.C. designed and performed research. A.B., M.C., analyzed data. L.P and D.K. provided mice, gave conceptual advice, and made corrections to the manuscript. P.G.H., A.B and M.C. wrote the paper.

## Competing interests

The authors declare no competing interests.
