## [Peer Review File · Nature Communications]

Astrocytic metabolic control of orexinergic activity in the lateral hypothalamus regulates sleep and wake architectureREVIEWER COMMENTS

Reviewer #1 (Remarks to the Author):

In their study, Braga and colleagues investigated the role of the astrocyte-neuron lactate shuttle (ANLS) in the sleep-wake architecture. Using pharmacological approaches, they found that lactate derived from astrocyte glycogen is essential for wakefulness stability. Using conditional knockouts of different lactate transporters in astrocytes or orexinergic neurons of the lateral hypothalamus, they also found in vivo that the astrocytic MCT4 and neuronal MCT2 transporters are involved in the shuttling of lactate between astrocytes and orexinergic neurons as well as in wakefulness stability. Using electrophysiological recordings in ex vivo hypothalamic brain slices, they also demonstrated that lactate shuttling via MCT4 and MCT2 is required to maintain orexinergic neuron firing.

This is a well-conceived, well-executed and well-written manuscript which provides complementary and compelling evidence for the critical role of the ANLS in regulating proper sleep/wake stability. These essential findings also provide another role for metabolic crosstalk between astrocytes and neurons and highlight the importance of glucose metabolites in various brain functions.

The authors used 500 μ M of 4-CIN in slices to block the neuronal lactate uptake. Such a relatively high concentration could also impair the mitochondrial uptake of pyruvate and subsequent ATP production and K-ATP channel closure, potentially leading to confounding effects. This possibility should be discussed.

The following minor points should be address to improve the quality of the manuscript:

Line 152: "NREM" should be defined here.

Fig. 1C: The choice of colors makes it difficult to see the aCSF condition in some radar charts (e.g. Light ZT0-12 /WAKE). Please modify the figure for clarity.

Extended data Fig. 1A and F.: Please make the "long homology arm", "LoxP flanked region" and "short homology arm" thicker. Currently they are hard to see. The forward genotyping primer appears to be placed on the 3' LoxP site of the MCT4 LoxP allele. Please, indicate where the genotyping primers are located on the WT allele. Similarly, indicate where the genotyping primers are located on the WT allele on the extended data Fig. 1F.

Extended data Fig. 1B legend: There is no "green" or "GFP" labeling in the Extended data Fig. 1B. Please also clarify the meaning of arrows. In the legend one should read: "GFAP in white (left panel) and Orexin-A (right panel)".

Line 321: Please quote a paper supporting the 2.5 mM physiological glucose concentration. (e.g. Silver and Erecinska, J. Neurosci. 1994).

Extended data Fig. 4A. Similar to Extended data Fig1A and F, it is unclear which region of the MCT2 gene is amplified in the WT allele.

Extended data Fig. 5A. The DAPI nuclear staining is barely visible.

Fig. 4: Since the recordings show spiking neurons, RMP (presumably standing for resting membrane potential) should be replaced by membrane potential (e.g. Vm).

Line 411: Extended data Fig. 7 is missing.

Line 432-433: One should read "...in oMCT2 KO mice (red, n=6) compared to control mice (black, n=5)..." not : "...or in oMCT2 KO mice compared to control (black, n=5) and oMCT2 KO mice (red, n=6)..."

Line 752: The unit of pipette resistance should be given in M Ω not in "MU".

Reviewer #2 (Remarks to the Author):

Summary: This manuscript explores the role of metabolic cooperation of lactate transport and uptake in vigilance states. In particular, how astrocytic transport of lactate by MCTs in the LV is necessary to maintain sleep/wake states. The authors use a series of complementary approaches to explore how astrocytes and orexinergic neurons work together to stabilize sleep. This paper is important because it provides a mechanistic explanation of how metabolic coupling by astrocyte-neuron is important for vigilance states and the stability of sleep/wake cycles. The authors use a variety of ex vivo and in vivo approaches to explore how MCT transport impacts sleep/wake via orexinergic neurons. While the premise of this paper is of great interest to the field, many of the conclusions the authors try to draw is not supported by their data or the data from one figure contradicts data from another figure. Some ways in which the authors present the data (e.g. radar plots) make it difficult for a reader to fully understand the results.

Criticisms:

- A major concern of this manuscript is the use of "KO" or "knockout" of either MCT2, MCT4, etc. The supplementary data suggests that these knockdowns are fairly minimal of total MCT expression/protein concentration. This is worth quantifying as well as discussing why a minimal knockout has a large effect.
- Major concern: please report Wake, NREM, REM activity for all graphs as your group has previously done in Figure 1 (Clasadone et al 2017, Neurons). This is an easier way to interpret your data than the radar plots. While radar plots have been used in the field, they are hard to read, do not show individual data points, and are unconvincing when reading author claims about significance and scale. Scales across axes are not consistent on these graphs, which makes it very hard to see differences across groups. Labeling of axes is also confusing. This reviewer suggests using "bout duration", "Bout number or # of bouts", and changing % recorded time to average minutes per hour or total minutes in recorded period. Bar graphs would be helpful to see spread of data, variance within groups, and to be able to visualize the degree of changes in sleep/wake across groups. It is also useful to the readers to be able to see how bout duration/number is contributing to overall sleep/wake time.
- It is unclear why the data during the light phase is binned in 12-hr bins while the dark phase is 6-hr bins. Since orexinergic neurons are important for transitions between arousal or vigilance states, it would be important to bin the light period in 6 hr bins as well.
- Extended data Figure 1 and Figure 2 seem contradictory. The extended data shows no difference in sleep/wake time at the beginning of the dark period because animals are awake for 5.5 hrs but then in the main figure show the opposite. The authors need to address this discrepancy.
- In the Results (page 4, line 163), the authors state the 4-CIN increased time spent in wakefulness during the light phase but decreased time in wakefulness during the dark phase. The authors need to explain the discrepancy between the light v dark phase data and how this fits into their model. Since mice do not have consolidated sleep, it is important to know whether orexinergic neurons, MCTs, or lactate could be doing something different to sleep/wake states based on time of day.
- The authors report wake, NREM, and REM as a percent time spent in that vigilant state. It would be beneficial to plot % time spent in each vigilant state per hour over the 24-hr period as they have done in previous manuscripts.
- Please be consistent on terminology with dark phase v active phase. Since there are changes in the vigilant states, it's better to describe the dark phase/period rather than use active phase since their time spent awake or active is shifting with the knockdown strategies employed.
- In Figure 1D, the authors report that blocking MCTs (lactate export and uptake) results in increased time spent awake in the light phase. They then claim in the dark phase, MCT blockage increases NREM. These results are contradictory. Naylor et al and others demonstrate that increased ISF lactate stimulates wakefulness across the 24-hour period, but this argues the opposite that less lactate flux is associated with increased time spent awake. This is incongruent with the authors findings. Can they explain?
- There is a lot of variance in sample size for sleep scoring. In some cases, it's an n=4 which

seems low given the variability of vigilant states. This is a potential confound on how to interpret the data.

- As stated above, we appreciate that radar plots are widely used in the field, however, they make the data hard to interpret. This is especially true with the changes in the axes between wake, NREM, and REM. It complicates the readers interpretation of the data. Moreover, with sample sizes equally $n=4$, it would be good to see individual data points to understand the distribution of scores. In extended figures, the variance of the data is high (e.g. Extended Figure 1).

- In extended figures, the authors quantify MCT4 and MCT1 knockdown using western blot. First, the percent of knockdown for both MCT4 and MCT1 seems minor where MCT4 is not statistically significant. What is the percent reduction in MCT1 or MCT4 protein levels? Looks less than 10% in MCT1 and highly variable in MCT4. This seems to be a major confound. Moreover, the authors should not refer to this as a KO or knockout throughout the manuscript if it is a low percentage knock down.

- The percent knockdown of each gene needs to be reported for each figure. Based on the extended figures, this is important to know given the phenotypes reported.

- Figure 3: Deletion of orexinergic MCT2. Authors report that control mice have bout durations of 12000s or close to 200min. This seems incorrect. Can the authors clarify?

- Figure 3D: Cag2 promoter is used in rescue experiments for MCT2. This will lead to overexpression in all cell types, not just neurons. This is a confound to their results and interpretation since many cell types (e.g. OPCs, oligodendrocytes, and microglia) also express MCT2. Moreover, rescue experiments must be compared to controls.

- Figure 3F – would like to see a comparison to controls, since bout duration actually looks higher than what it is in controls, even if it is just a supplemental figure.

- It is unclear what "duration" means in radar plots. They vary from 6000-12000 in controls. Is this total sleep/wake time? Bout duration? Please clarify.

- Figure 4: % of OrexinA positive neurons was quantified. It is important to see that MCT2 and MCT4 are present and depleted with western blot or immunostaining.

- Figure 4: It's interesting that extracellular lactate doesn't increase firing in control cells. Can the authors speculate to why it doesn't? It might be important to quantify lactate levels in the bath under basal conditions in control v mct4 ko. This would confirm that lactate shuttling is impaired in the mct4ko mice and basal lactate levels differ between groups in the presence of 2.5mM glucose.

- Figure 4: Figures are referred to as C1, C2, etc. It would be better for each panel to have its own letter assigned to it.

- Figure 4 E and H: show sample traces for controls, since in 4G there isn't an increasing rate of firing with lactate or Tolbutamide administration in controls.

- Line 226-228 states that MCT1KO mice are compared to MCT4KO but they are not (see extended figure 3), they are compared to control.

- Line 367: "These results suggest that lactate, shuttled by astrocytes through MCT4, needs to be internalized through MCT2 for sustaining orexinergic excitability through KATP channel gating." This is a strong statement that needs to be modified. The data demonstrates that lactate shuttling is necessary for orexinergic excitability and can be rescued by stimulating KATP channel activity. The relationship between lactate and KATP channels in this instance is correlative.

- Figure 5: In the MCT4KO mice, there is a reduction in lactate compared to controls. The authors showed previously in Figure 4 that firing rate is reduced. This may be a circular argument but if firing rate is reduced then it reasons that the stimulation of lactate production would be reduced according to the ANLS. Can the authors normalize this to total EEG power?

- Figure 5B & D: These plots are very confusing. What is Wake Rel. Freq.? Is it power? It is unclear whether these graphs are necessary or what they are showing. The controls in B and D look completely different in the frequency distribution of lactate so it's unclear what this is demonstrating. Moreover, the control curves in C and E look significantly different from one another. Why is the change in lactate this variable between controls?

- If orexinergic neurons are responsive to lactate per the discussion then why does lactate not increase firing in control mice? This is central to the authors hypotheses and needs to be demonstrated.

- Line 513: Authors claim that they observed daily fluctuations in extracellular lactate and it relates to changes in sleep; however, this data is not shown. Please include this.

- Line 553: authors posit that neurons express functional KATP channels composed of Kir6.1 and Sur1 and cite several papers. First, most neuronal KATP channels are composed of Kir6.2, not Kir6.1 (Grizzanti et al 2023). Kir6.1 is thought to be more vascular in origin (see Vine-seq single

cell database, Seattle AD brain atlas, Brainseq.com for expression levels). The citations list (20,21,51,52) do not demonstrate Kir6.1 is on neurons but rather demonstrate that KATP channel activity is important for LH or orexinergic neurons.

- Line 411: references extended figure 7 which does not exist. Please remove.

Reviewer #3 (Remarks to the Author):

Neurons need energy fuel to work. A large body of evidence, reviewed well by the authors, indicates that in the brain an important component of this fuel comes from astrocyte-derived lactate, rather than directly from glucose.

The authors (correctly) point out that the importance of this "astrocyte-neuron lactate shuttle" (ANLS) for neural activity is already well established in very many brain areas.

This specific paper elucidates the role of astrocyte-neuron lactate shuttle (ANLS) in providing fuel to lateral hypothalamic orexin neurons, whose activity is already well established (by work over the past 25 years) to be essential for maintaining wakefulness.

The authors perform standard experiments to interfere with various elements of ANLS, such as MCT transporters, and to measure lactate in the lateral hypothalamus where orexin neurons are found. As expected from the many previously published findings about ANLS importance for maintaining neural firing across the brain, and about the importance of orexin cells for wakefulness, the authors find that interference with LH ANLS compromises activity of orexin cells and thus compromises wakefulness control.

On a technical level, I find the study very well done. The authors have been leaders and pioneers in the ANLS field for many years, and they apply their standard tools well.

However, I find most of the findings very confirmatory and not surprising. The authors showed that yet another brain area relies on ANLS, and confirm that neurons cannot function properly without fuel. I would therefore appreciate more explanation, in introduction and discussion, by what major "gap in knowledge" is filled by this paper. Without this, the study comes across as confirmatory and unsurprising.

Reviewer #4 (Remarks to the Author):

This is a very interesting study exploring the role of astrocytic support to neurons in the modulation of sleep and wake. This builds on earlier work from the authors showing that astrocytic lactate influences the activity of neurons important in these processes. Here they use similar approaches to examine orexin/hypocretin+ (ORX+) neurons in the lateral hypothalamus. There are several strengths, including a multi-platform approach ranging from in vivo to in vitro, transgenic/viral approaches, lactate sensors and patch-clamp electrophysiology. There are, however, some issues that need to be addressed to improve the impact and clarity of the results.

1. In several places results/interpretations reported in the text or legends do not appear to match what is shown in the figures. This in part may be due to the non-standard way of presenting the sleep & wake data ("radar" plots, see below). For example, the amct4 ko mice are reported in the text to have reductions in wake and nrem sleep time in the first half of the dark phase, but that is not what is shown in Figure 2C: wake as % Rec. time is decreased, and nrem % is increased, which makes sense—as REM % does not change much. Legend in Figure 2, "decreased duration of wakefulness and NREM epochs" is reported. How can a 4-second epoch be decreased in duration? This is confusing. Also, not what I see. I see a reduction in wake and NREM % rec. time and no change in NREM "duration". This may reflect a problem in how terms are being used and defined (see point 2).

2. There are inconsistencies in how the three main measures of sleep & wake are described (% recording time, duration, and # of events). For example, sometimes the % recording time data are

referred to as “duration of time” (line 217), but duration appears to refer to how long an “event” is. BTW, what is an “event”? I assume this is different than a 4 second epoch? If it is a measure of how long an individual occurrence of a state is (more commonly referred to as a “bout”), then the minimum length of a bout needs to be defined. This can range widely from study to study (e.g., 4, 8, 16, 20 seconds). Also, in other places, “episodes” are reported. Is this different than an “event”? Please go through the MS and use consistent terminology (once defined).

3. The radar plots are hard to interpret, especially when lines and symbols overlap and plots are so small. For example, Fig 3c, changes in wake as % rec. time have symbols that are overlapping, yet this is reported as the most significant change in wake (***, although not defined in the legend, I assume 3* are > than 2*). In the same plot, oMCT2 KO wake duration appears to be at the origin, while the controls are around 6000 seconds—this is not significant? Although I see the advantage of one plot summarizing a lot of data, this is a non-standard way of presenting sleep & wake data. This makes it difficult to compare to earlier work using more common displays. I don’t see much advantage to doing it this way—and I note the authors used more standard histograms for sleep & wake data in the extended figures. I strongly recommend they use bar histograms with SEMs for the sleep & wake data in the main figures.

4. I have several questions about the oMCT2 KO mice data. The representative hypnogram data shown in Figure 3 are internally inconsistent and puzzling. First of all, panel B shows a near complete disorganization of sleep and wake organization and a huge increase in REM sleep in the KO relative to control. This looks like a narcoleptic mouse—which itself could be a very interesting finding, as there is only a trend for a loss of ORX+ neurons from this embryonic KO. Yet, in panel E, another KO mouse shows a distribution of sleep & wake that more closely resembles the control mouse in panel B! While on average, the data may support the author’s interpretations, these representative data do not. Also, as a discussion point, why didn’t the authors use a viral based approach to KO MCT2 in ORX+ neurons in adults? Isn’t it possible that embryonic KO might result in perturbations in ORX+ neurons that go beyond adult cell use of lactate? This might explain why changes in adults differ in some ways from the other manipulations.

5. The authors conclude that astrocytic lactate support is a “major regulator” of sleep/wake architecture or is “necessary for maintaining wakefulness during ZT12-18”. These statements are too strong and not supported by the data. The effects are significant, but modest in magnitude and restricted in time. I also wonder if their results might be further restricted to only the first hour or 2 after lights out. If so, this would further circumscribe these results (and raise questions about circadian factors)—but there is no way to know, as hourly values are not shown. Only 6 & 12 hour averages. Moreover, there is no discussion at all about why, if this is a major regulator of sleep & wake, this is restricted to the first half of the dark phase? As a major regulator, one would expect similar effects at all ZT times.

6. The authors should really address caveats in the interpretation of their results—especially as the discussion is too long; it should be shortened to accommodate a discussion of caveats. The different manipulations produce similar, but also different results (e.g, 4-CIN vs. MCT4 KO vs. oMCT2 KO), and this is likely due to the strengths and weaknesses of each approach. For example, the minipump infusions are into the LH, which contain neurons other than ORX+. So, how far is the effective infusion zone? What other neurons are there that can directly or indirectly impact sleep & wake architecture? Do these manipulations impact neurons that regulate core temperature, motor activity or pain? See above for points regarding the oMCT2 KO mice.

Additional points

1. male and female mice were used. Were there any sex differences?
2. For the lactate sensor data, only EEG signals were used to score REM, Wake and NREM? How was this validated? cortical EEG signals are not the best means of scoring wake vs REM.
3. Did any of these manipulations effect core temperature?

Reviewer #5 (Remarks to the Author):

We would like to thank all the reviewers for dedicating their time to evaluate our submission and the very positive assessment of our work. We appreciate the detailed and constructive comments provided. We are thankful for the opportunity to submit a revised version of our work. In this letter, we address all the criticisms raised comprehensively and present a thoroughly revised manuscript.

Reviewer #1 Remarks to the author:

In their study, Braga and colleagues investigated the role of the astrocyte-neuron lactate shuttle (ANLS) in the sleep-wake architecture. Using pharmacological approaches, they found that lactate derived from astrocyte glycogen is essential for wakefulness stability. Using conditional knockouts of different lactate transporters in astrocytes or orexinergic neurons of the lateral hypothalamus, they also found in vivo that the astrocytic MCT4 and neuronal MCT2 transporters are involved in the shuttling of lactate between astrocytes and orexinergic neurons as well as in wakefulness stability. Using electrophysiological recordings in ex vivo hypothalamic brain slices, they also demonstrated that lactate shuttling via MCT4 and MCT2 is required to maintain orexinergic neuron firing.

This is a well-conceived, well-executed and well-written manuscript which provides complementary and compelling evidence for the critical role of the ANLS in regulating proper sleep/wake stability. These essential findings also provide another role for metabolic crosstalk between astrocytes and neurons and highlight the importance of glucose metabolites in various brain functions.

The authors used 500 μ M of 4-CIN in slices to block the neuronal lactate uptake. Such a relatively high concentration could also impair the mitochondrial uptake of pyruvate and subsequent ATP production and K-ATP channel closure, potentially leading to confounding effects. This possibility should be discussed.

We agree with the reviewer, and we cannot completely exclude the effect of 4-CIN on other targets, such as the mitochondrial pyruvate carrier. In our case, however, the application times were less than 20 minutes, resulting in incomplete equilibration of 4-CIN in the slice PMID: [27559140](https://pubmed.ncbi.nlm.nih.gov/27559140/). Thus, it is less expected that the applied concentration will act on mitochondrial pyruvate uptake.

Indeed, to act on the mitochondrial pyruvate carrier, 4-CIN must first enter the cell via MCT2 and accumulate, requiring more than 20 minutes. To further demonstrate the key role of MCT2, we deleted MCT2 in orexinergic neurons (Fig. 4D). In this case, we observed decreased neuronal activity not rescued by extracellular lactate delivery, supporting a key role for MCT2 mediated lactate uptake. We included the limitations of 4-CIN in the revised version of the manuscript by stating: "it is noteworthy that we cannot exclude off-target effects of 4-CIN, including extracellular and intracellular acidification due to impaired proton transport, inhibition of mitochondrial pyruvate transport, or effects on other cell types, rather than astrocytes expressing MCTs." (line 522-525)

The following minor points should be addressed to improve the quality of the manuscript:

Line 152: "NREM" should be defined here.

We thank the reviewer for the suggestion. We have now edited the text accordingly.

Fig. 1C: The choice of colors makes it difficult to see the aCSF condition in some radar charts (e.g. Light ZT0-12 /WAKE). Please modify the figure for clarity.

We thank the reviewer for the comment. The graphs have now been changed to reflect suggestions from other reviewers. Radar charts have been replaced with bar graphs and color distinctions should now be clearer.

Extended data Fig. 1A and F.: Please make the "long homology arm", "LoxP flanked region" and "short homology arm" thicker. Currently they are hard to see. The forward genotyping primer appears to be placed on the 3' LoxP site of the MCT4 LoxP allele. Please, indicate where the genotyping primers are located on the WT allele. Similarly, indicate where the genotyping primers are located on the WT allele on the extended data Fig. 1F. We thank the reviewer for the comment, the figure has been changed accordingly.

Extended data Fig. 1B legend: There is no "green" or "GFP" labeling in the Extended data Fig. 1B. Please also clarify the meaning of arrows. In the legend one should read: "GFAP in white (left panel) and Orexin-A (right panel)".

We thank the reviewer for identifying the missing GFP image. The arrows indicate representative GFAP and Td-tomato double-positive astrocytes and Orexin-A and Td-Tomato double-positive orexinergic neurons, as quantified in the bar graphs next to the representative images.

Line 321: Please quote a paper supporting the 2.5 mM physiological glucose concentration. (e.g. Silver and Erecinska, J. Neurosci. 1994)

We thank the reviewer for the suggestion, the reference has been added to the text.

Extended data Fig. 4A. Similar to Extended data Fig1A and F, it is unclear which region of the MCT2 gene is amplified in the WT allele.

We thank the reviewer for the suggestion, the figure has been corrected and implemented for clarity.

Extended data Fig. 5A. The DAPI nuclear staining is barely visible.

We thank the reviewer for the suggestion. Dapi was enhanced with the linear contrast enhancement function of ImageJ.

Fig. 4: Since the recordings show spiking neurons, RMP (presumably standing for resting membrane potential) should be replaced by membrane potential (e.g. Vm).

We thank the reviewer for the suggestion the figure has been changed accordingly.

Line 411: Extended data Fig. 7 is missing.

We have now corrected the text.

Line 432-433: One should read "...in oMCT2 KO mice (red, n=6) compared to control mice (black, n=5)..." not : "...or in oMCT2 KO mice compared to control (black, n=5) and oMCT2 KO mice (red, n=6)..."

We thank the reviewer for the correction. We have adjusted the text accordingly.

Line 752: The unit of pipette resistance should be given in M Ω not in "MU".

We thank the reviewer for the correction. We have adjusted the text accordingly

Reviewer #2 (Remarks to the Author):

Summary: This manuscript explores the role of metabolic cooperation of lactate transport and uptake in vigilance states. In particular, how astrocytic transport of lactate by MCTs in the LV is necessary to maintain sleep/wake states. The authors use a series of complementary approaches to explore how astrocytes and orexinergic neurons work together to stabilize sleep. This paper is important because it provides a mechanistic explanation of how metabolic coupling by astrocyte-neuron is important for vigilance states and the stability of sleep/wake cycles. The authors use a variety of ex vivo and in vivo approaches to explore how MCT transport impacts sleep/wake via orexinergic neurons. While the premise of this paper is of great interest to the field, many of the conclusions the authors try to draw is not supported by their data or the data from one figure contradicts data from another figure. Some ways in which the authors present the data (e.g. radar plots) make it difficult for a reader to fully understand the results.

Criticisms:

- A major concern of this manuscript is the use of "KO" or "knockout" of either MCT2, MCT4, etc. The supplementary data suggests that these knockdowns are fairly minimal of total MCT expression/protein concentration. This is worth quantifying as well as discussing why a minimal knockout has a large effect.

We agree with the reviewer that 'knockout' is not the appropriate term; therefore, we have revised it to 'knockdown mice' in the manuscript. In terms of assessing the knockdown efficiency, we performed the isolation of the lateral hypothalamus and extracted proteins for quantification. This method encompasses all cell types within the area. Although MCT4 expression is anticipated to be predominantly in astrocytes, a recent study suggested that microglial cells also express MCT4 (PMID: 37717033). Thus, a reduction of about 50% is expected. Likewise, MCT1 is expressed by various cell types in the brain, including oligodendrocytes and endothelial cells (PMID: 15953344). We have added the percentage of reduction measured in our Western Blots and addressed this limitation in the Materials and Methods by stating "Due to the confined size of the lateral hypothalamus, we quantified protein downregulation from the entire isolated area, thus including all cell types within the tissue."

- Major concern: please report Wake, NREM, REM activity for all graphs as your group has previously done in Figure 1 (Clasadonte et al 2017, Neurons). This is an easier way to interpret your data than the radar plots. While radar plots have been used in the field, they are hard to read, do not show individual data points, and are unconvincing when reading author claims about significance and scale. Scales across axes are not consistent on these graphs, which makes it very hard to see differences across groups. Labeling of axes is also confusing. This reviewer suggests using "bout duration", "Bout number or # of bouts", and changing % recorded time to average minutes per hour or total minutes in recorded period. Bar graphs would be helpful to see spread of data, variance within groups, and to be able to visualize the degree of changes in sleep/wake across groups. It is also useful to the readers to be able to see how bout duration/number is contributing to overall sleep/wake time.

We appreciate the reviewer's input and have addressed the comment by replacing the radar plots with bar graphs. For the y-axis nomenclature, we have retained "Episode number" and "Average episode duration (s)" to maintain consistency with our prior work (Clasadonte et al., 2017, Neuron), ensuring clarity for a broader audience. Moreover, we believe that expressing "% recording time" accurately represents the data in the graph, as these values are calculated as a percentage of time rather than in minutes. Nevertheless, we have also added in the Supplementary figures the % of each stage (wake, NREM and REM) per hour to simplify visualization of sleep/wake cycle across experiments.

- It is unclear why the data during the light phase is binned in 12-hr bins while the dark phase is 6-hr bins. Since orexinergic neurons are important for transitions between arousal or vigilance states, it would be important to bin the light period in 6 hr bins as well.

We opted to divide only the dark phase into 6-hour bins based on evidence showing that orexinergic neurons play a crucial role in sustaining prolonged wakefulness during this phase (dark phase) (Clasadonte et al., 2017, Neuron). Moreover, previous studies employing Orexin knockout mice (mice lacking either orexin expressing neurons or orexin receptors, PMID: 12797957; PMID: 15254084) demonstrated the significance of orexinergic neurons in maintaining wakefulness during the dark phase, while their impact during the light phase is minimal to none. Transitions from sleep to wake are likely not significantly impaired, considering that other elements important in sleep/wake transition, such as the locus coeruleus, remain intact and can still promote such behavior (PMID: 21037585). In the Supplementary figures provided in this revised version (Fig. S1, S3 and S5) it is further clear that manipulation does not affect the light phase and has a major impact in the first part of the dark phase.

- Extended data Figure 1 and Figure 2 seem contradictory. The extended data shows no difference in sleep/wake time at the beginning of the dark period because animals are awake for 5.5 hrs but then in the main figure show the opposite. The authors need to address this discrepancy.

We apologize for the confusion. Supplementary Figure 1 (in the current manuscript revision Supplementary Figure 2) refers to data collected 6 weeks after the viral injections, where no protein reduction was observed. Figure 2 refers to data collected 12 weeks after viral injections, when we observed a reduction in protein expression. We have further clarified the two time points in the Results section.

- In the Results (page 4, line 163), the authors state the 4-CIN increased time spent in wakefulness during the light phase but decreased time in wakefulness during the dark phase. The authors need to explain the discrepancy between the light v dark phase data and how this fits into their model. Since mice do not have consolidated sleep, it is important to know whether orexinergic neurons, MCTs, or lactate could be doing something different to sleep/wake states based on time of day.

We observed a difference mean of 3.506 for wakefulness during the light phase whereas a more substantial mean difference of 11.96 was observed for wakefulness during the dark phase. Additionally, the effect sizes varied, with the light phase showing a smaller effect size of 0.608 compared to a larger effect size of 0.96 during the dark phase. As 4-CIN is a non-selective MCTs inhibitor and various brain cells express different MCT isoforms (microglia PMID: 37717033, oligodendrocyte PMID: 33440165, endothelial cells PMID: 16403470), we cannot exclude the possibility that our pharmacological approach has targeted these cells and influenced their behavior during the light phase. Notably, the increase of wakefulness during the light is indeed specific to 4-CIN application as blocking glycogenolysis (a process mainly astrocytic) had no effect during the light phase, resembling our astrocytic MCT4 knockdown mice. We have addressed these off-target pharmacological effects in our revised manuscript in the Discussion section by stating: "We also detected an effect during the light phase, and it is noteworthy that we cannot exclude off-target effects of 4-CIN, including extracellular and intracellular acidification due to impaired proton transport (PMCID: PMC1965573), inhibition of mitochondrial pyruvate transport (PMID: 11746398), or effects on other cell types, rather than astrocytes expressing MCTs."

- The authors report wake, NREM, and REM as a percent time spent in that vigilant state. It would be beneficial to plot % time spent in each vigilant state per hour over the 24-hr period as they have done in previous manuscripts.

We have added % time spent in each state per hour over the 24h period in the Supplementary Figures (Supplementary Figure 1, 3 and 5).

- Please be consistent on terminology with dark phase v active phase. Since there are changes in the vigilant states, it's better to describe the dark phase/period rather than use active phase since their time spent awake or active is shifting with the knockdown strategies employed.

In the revised manuscript, we have rectified the terminology to ensure a more consistent use of the term "dark phase."

- In Figure 1D, the authors report that blocking MCTs (lactate export and uptake) results in increased time spent awake in the light phase. They then claim in the dark phase, MCT

blockage increases NREM. These results are contradictory. Naylor et al and others demonstrate that increased ISF lactate stimulates wakefulness across the 24-hour period, but this argues the opposite that less lactate flux is associated with increased time spent awake. This is incongruent with the authors findings. Can they explain?

As elucidated in the previous response, 4-CIN is a non-selective MCT inhibitor, and due to the widespread expression of MCT isoforms across various cell types in the brain, the involvement of other cells, such as oligodendrocytes, microglia, or endothelial cells, cannot be excluded. It's crucial to clarify that Naylor et al. (PMID: 22942499) did not perform interventions involving lactate infusion or application; instead, they measured changes in extracellular lactate in the prefrontal cortex of C57Bl/6J mice. While their results indicated a correlation between extracellular lactate and wakefulness (Figure 6 of their manuscript), they did not demonstrate how the manipulation of extracellular lactate levels could alter wakefulness, particularly in the lateral hypothalamus, as demonstrated in our manuscript.

- There is a lot of variance in sample size for sleep scoring. In some cases, it's an n=4 which seems low given the variability of vigilant states. This is a potential cofound on how to interpret the data.

We acknowledge the variance in sample size. However, for within-subject experiments, we choose to limit the number of mice. We have now replaced the radar graph with bar plots to increase the clarity in variability.

- As stated above, we appreciate that radar plots are widely used in the field, however, they make the data hard to interpret. This is especially true with the changes in the axes between wake, NREM, and REM. It complicates the readers interpretation of the data. Moreover, with sample sizes equally n=4, it would be good to see individual data points to understand the distribution of scores. In extended figures, the variance of the data is high (e.g. Extended Figure 1)

We appreciate the reviewer's input and have addressed the comment by replacing the radar plots with bar graphs

- In extended figures, the authors quantify MCT4 and MCT1 knockdown using western blot. First, the percent of knockout down for both MCT4 and MCT1 seems minor where MCT4 is not statistically significant. What is the percent reduction in MCT1 or MCT4 protein levels? Looks less than 10% in MCT1 and highly variable in MCT4. This seems to be a major confound. Moreover, the authors should not refer to this as a KO or knockout throughout the manuscript if it is a low percentage knock down.

We agree with the reviewer that knockout is not the appropriate term, we have revised this term to "knockdown mice" in the revised manuscript. In terms of assessing the knockdown efficiency, we performed isolation of the lateral hypothalamus and extracted proteins for quantification given the very small area and the difficulties in achieving a meaningful yield of astrocytes by other means, such as Fluorescence Activated Cell Sorting. This method encompasses all cell types within the area. Although MCT4 expression is anticipated to be predominantly in astrocytes, a recent study suggested that also microglia express MCT4 (PMID: 37717033). Thus a reduction of about 50% is expected. Likewise, MCT1 is highly expressed by various cell types in the brain

(PMID:15953344). We have addressed this limitation in the Material and Methods section by stating “We also detected an effect during the light phase, it is noteworthy that we cannot exclude off-target effects of 4-CIN, including extracellular and intracellular acidification due to impaired proton transport, inhibition of mitochondrial pyruvate transport, or effects on other cell types, rather than astrocytes expressing MCTs.”

- The percent knockdown of each gene needs to be reported for each figure. Based on the extended figures, this is important to know given the phenotypes reported.

We agree in principle with the reviewer. Nevertheless protein quantification for each cohort in every experiment would have not been feasible as tissues were also used for other experiments (such as PCR or staining). We ensured that Western blot reflected the mice population used in our experiments by using tissue from mice previously employed for EEG and/or electrophysiological recordings. In mice that were not used for Western blot, after each experiment, we verified the expression of GFP in the lateral hypothalamus as assessment for viral expression and to confirm the expression of Cre recombinase and as a proxy for protein knockdown.

- Figure 3: Deletion of orexigenic MCT2. Authors report that control mice have bout durations of 12000s or close to 200min. This seems incorrect. Can the authors clarify? In the revised manuscript, we have included bar graphs illustrating that, in every control sample throughout the study, certain mice exhibited notably extended periods of wakefulness, contributing to the elevated calculated average.

- Figure 3D: Cag2 promoter is used in rescue experiments for MCT2. This will lead to overexpression in all cell types, not just neurons. This is a confound to their results and interpretation since many cells types (e.g. opcs, oligodendrocytes, and microglia) also express MCT2. Moreover, rescue experiments must be compared to control s.

- Figure 3F – would like to see a comparison to controls, since bout duration actually looks higher than what it is in controls, even if it is just a supplemental figure.

Response for comment to Figure 3D and 3F: For this experiment we took advantage of our Orexin-Cre line and used a Cre-dependent viral approach. Despite the generic Cag2 promoter the expression of Slc16a7 is controlled by the DIO (double-floxed inverse open reading frame) system limiting its expression to orexin-cre expressing neurons. In our experimental design, our initial focus was on evaluating the impact of MCT2 knockdown on sleep and wake cycle (observing a reduction in wakefulness in oMCT2 KO). Subsequently, our objective shifted to investigating whether the reintroduction of MCT2 in orexinergic neurons could enhance wakefulness. Hence, we opted for a within-subject approach, not only for its practicality in minimizing the number of mice utilized but also for its alignment with our experimental objectives. We acknowledge that a comparison with a control group could provide insights into whether re-expressing MCT2 restores wakefulness to a physiological level. However, given our specific aim of enhancing wakefulness, a paired experimental design satisfied our investigative purposes.

- It is unclear what “duration” means in radar plots. They vary from 6000-12000 in controls. Is this total sleep/wake time? Bout duration? Please clarify.

In the revised manuscript we have changed the radar plots to bar graphs and changed the y-axis labeling to average episode duration (s). It represents the average duration of wake, NREM or REM episodes.

- Figure 4: % of OrexinA positive neurons was quantified. It is important to see that MCT2 and MCT4 are present and depleted with western blot or immunostaining.

We agree with the reviewer's comment; however, there are some technical limitations to this experiment. We did not perform WB on oMCT2 KO because orexinergic neurons constitute a tiny population in the mouse hypothalamus, approximately 5,000 neurons. Therefore, isolating enough tissues to yield reliable WB results was challenging. Additionally, for the immunostaining, the antibody against MCT2 did not give reliable results in our hands, showing non-specific signals. To overcome this technical limitation and demonstrate that the sleep phenotype observed in oMCT2 KO was due to the lack of this transporter in these neurons, we employed a viral strategy to re-express MCT2 in orexinergic neurons from oMCT2KO mice. These results confirmed that MCT2 is fundamental and required to sustain wakefulness during the dark phase.

- Figure 4: It's interesting that extracellular lactate doesn't increase firing in control cells. Can the authors speculate to why it doesn't? It might be important to quantify lactate levels in the bath under basal conditions in control v mct4 ko. This would confirm that lactate shuttling is impaired in the mct4ko mice and basal lactate levels differ between groups in the presence of 2.5mM glucose.

To support our in vivo data (Fig. 5) and address this question, we conducted whole-cell recordings, manipulating lactate concentrations (1, 2.5, 5, and 10 mM) in the absence of glucose. We measured the firing rate of orexinergic neurons as an output. Notably, we observed a concentration-dependent increase in firing rate. Interestingly increasing the concentration from 5 to 10-mM lactate did not further change the firing rate, implying a ceiling effect on neuronal activity. Furthermore, 10- and 5-mM lactate displayed a firing rate comparable to that of 2.5 mM glucose. This suggests that the endogenous lactate level reached a plateau at a glucose concentration of 2.5 mM in our hypothalamic brain slices. Consequently, elevating lactate concentration in the presence of 2.5 mM glucose did not impact the firing rate of control cells. We include these new results in Supplementary Figure 8 d.

- Figure 4: Figures are referred to as C1, C2, etc. It would be better for each panel to have its own letter assigned to it.

We revised the figure and assigned a letter for each panel. We changed figure legends accordingly.

- Figure 4 E and H: show sample traces for controls, since in 4G there isn't an increasing rate of firing with lactate or Tolbutamide administration in controls.

In the revised version of the manuscript, we added the missing control traces in Supplementary Figure 8 a, b, c.

- Line 226-228 states that MCT1KO mice are compared to MCT4KO but they are not (see extended figure 3), they are compared to control.

We thank the reviewer for the comment, and we agree. We have now removed it from the revised manuscript.

- Line 367: "These results suggest that lactate, shuttled by astrocytes through MCT4, needs to be internalized through MCT2 for sustaining orexinergic excitability through KATP channel gating." This is a strong statement that needs to be modified. The data demonstrates that lactate shuttling is necessary for orexinergic excitability and can be rescued by stimulating KATP channel activity. The relationship between lactate and KATP channels in this instance is correlative. – In the revised manuscript we adjusted the wording by stating: "the activity of these neurons in oMCT2 KD and aMCT4 KD mice can be rescued by inhibiting K_{ATP} channel activity." (line 396-397)

Figure 5: In the MCT4KO mice, there is a reduction in lactate compared to controls. The authors showed previously in Figure 4 that firing rate is reduced. This may be a circular argument but if firing rate is reduced then it reasons that the stimulation of lactate production would be reduced according to the ANLS. Can the authors normalize this to total EEG power?

Based on this assumption we should be observing reduced extracellular lactate *in vivo* in oMCT2 KD given their reduced neuronal activity. However, the opposite is observed, in support of the source and sink relationship that we discuss. Normalizing on the EEG power might not be entirely accurate since EEG is derived from whole-brain signals, and using local LFP (Local Field Potential) could be more indicative for such purposes. Unfortunately, technical constraints, such as the presence of the lactate probe, make it unfeasible to incorporate an electrode for local LFP measurements.

- Figure 5B & D: These plots are very confusing. What is Wake Rel. Freq.? Is it power? It is unclear whether these graphs are necessary or what they are showing. The controls in B and D look completely different in the frequency distribution of lactate so it's unclear what this is demonstrating. Moreover, the control curves in C and E look significantly different from one another. Why is the change in lactate this variable between controls?

In the revised manuscript we removed the Frequency distribution of lactate concentrations during wakefulness in ZT12-28 and showed only the cumulative distributions of lactate concentration. The latter is derived from the frequency distribution. Data from control mice in the MCT4 and MCT2 groups were collected at different ages due to the experimental design. The time required to observe a reduction in MCT4 expression (12 weeks after injection) results in these mice (and their controls) being older than those in the MCT2 group, potentially accounting for observed age-related differences in lactate (PMID: 29678731).

- If orexinergic neurons are responsive to lactate per the discussion then why does

lactate not increase firing in control mice? This is central to the authors hypotheses and needs to be demonstrated. Ceiling effect.

Thanks for this comment. We have added a new experiment demonstrating a ceiling effect on orexinergic activity at a concentration of 5 mM lactate in the absence of glucose. Increasing the concentration to 10 mM does not further increase the firing rate. Furthermore, 10 mM and 5 mM lactate displayed a firing rate comparable to that of 2.5 mM glucose. We included these new results in Supplementary Fig. 8d.

In supporting of these findings, our data also showed that tolbutamide, an inhibitor of the K_{ATP} channel, did not further increase the firing rate of control cells but was able to increase the activity of orexinergic neurons from aMCT4KD and oMCT2KD mice (Fig. 4m, n and Supplementary Fig. 8d).

These data collectively suggest that the metabolism-sensing machinery involving K_{ATP} channels is saturated at 2.5 mM glucose in our control brain slices preparation. The endogenous lactate production in control cells is sufficient to keep K_{ATP} channels in a closed configuration, maintaining the high frequency of orexinergic neurons.

- Line 513: Authors claim that they observed daily fluctuations in extracellular lactate and it relates to changes in sleep; however, this data is not shown. Please include this.

We have now added the representative traces for lactate fluctuation in sleep and wake in Fig.5b.

- Line 553: authors posit that neurons express functional K_{ATP} channels composed of Kir6.1 and Sur1 and cite several papers. First, most neuronal K_{ATP} channels are composed of Kir6.2, not Kir6.1 (Grizzanti et al 2023). Kir6.1 is thought to be more vascular in origin (see Vine-seq single cell database, Seattle AD brain atlas, Brainseq.com for expression levels). The citations list (20,21,51,52) do not demonstrate Kir6.1 is on neurons but rather demonstrate that K_{ATP} channel activity is important for LH or orexinergic neurons.

We thank the reviewer for this comment however we based this affirmation on Figure 4 of Matthew P. Parsons and Michiru Hirasawa's manuscript (PMID 20554857). The figure shows colocalization between Kir6.1 and orexinergic neurons but not Kir 6.2. We have now relocated references to correspond with the text.

- Line 411: references extended figure 7 which does not exist. Please remove.

We have now corrected the text.

Reviewer #3 (Remarks to the Author):

Neurons need energy fuel to work. A large body of evidence, reviewed well by the authors, indicates that in the brain an important component of this fuel comes from astrocyte-derived lactate, rather than directly from glucose.

The authors (correctly) point out that the importance of this “astrocyte-neuron lactate shuttle” (ANLS) for neural activity is already well established in very many brain areas. This specific paper elucidates the role of astrocyte-neuron lactate shuttle (ANLS) in

providing fuel to lateral hypothalamic orexin neurons, whose activity is already well established (by work over the past 25 years) to be essential for maintaining wakefulness.

The authors perform standard experiments to interfere with various elements of ANLS, such as MCT transporters, and to measure lactate in the lateral hypothalamus where orexin neurons are found. As expected from the many previously published findings about ANLS importance for maintaining neural firing across the brain, and about the importance of orexin cells for wakefulness, the authors find that interference with LH ANLS compromises activity of orexin cells and thus compromises wakefulness control.

On a technical level, I find the study very well done. The authors have been leaders and pioneers in the ANLS field for many years, and they apply their standard tools well.

However, I find most of the findings very confirmatory and not surprising. The authors showed that yet another brain area relies on ANLS, and confirmed that neurons cannot function properly without fuel. I would therefore appreciate more explanation, in introduction and discussion, by what major "gap in knowledge" is filled by this paper. Without this, the study comes across as confirmatory and unsurprising.

In the following points, we clarify the innovation of the manuscript:

-We are reporting a significant role for astrocytic MCT4 in the consolidation of wakefulness. Our data indicate that astrocytic MCT4, but not MCT1, is necessary for consolidating wakefulness during the dark phase. Considering that alterations in MCT4 expression have been documented in aging (PMID: 24244584) and in both patients and murine models of Alzheimer's disease (PMID: 31738978) and recognizing that sleep disturbances are common in aging and AD, this discovery opens up new possibilities. With this information, it may become feasible to use modulators of MCT4 to control the sleep/wake cycle and alleviate sleep/wake-related phenotypes, thereby enhancing the quality of life for patients.

-Whether lactate needs to be internalized to support neuronal activity is still a matter of controversy. Some studies propose that neurons do not import lactate but they export lactate, increasing their glycolytic rate PMID: 28768175, while others suggest that lactate functions via an extracellular G-coupled receptor PMID: 30926749. In this study, we crossed Orexin-IRES-Cre mice and MCT2 f/f mice to establish a knockdown model of the main importer of lactate (oMCT2 KD mice). Through the use of a lactate biosensor implanted in the LH, our findings reveal for the first time that the deletion of orexinergic MCT2, the neuronal lactate importer, results in the accumulation of extracellular lactate in the extracellular space. This supports our assertion, presented throughout the paper, that orexinergic neurons act as lactate sinks.

-Finally, despite the physiological concentration of glucose in our *in vitro* and *in vivo* experiments, reducing the lactate shuttle resulted in an alteration of sleep/wake patterns and orexinergic activity. It is noteworthy that in our experiments, orexinergic neurons can

still import glucose as energy source. This suggests that glucose and lactate are not interchangeable in modulating orexinergic activity and wakefulness. We propose that lactate is a fundamental fuel for the first 6 hours of the dark phase. This is possibly because orexinergic neurons require an extra and readily available boost of energy to support the transition from sleep to wake. This makes lactate the metabolic choice of preference for high-demanding tasks such as prolonged wakefulness.

Reviewer #4 (Remarks to the Author):

This is a very interesting study exploring the role of astrocytic support to neurons in the modulation of sleep and wake. This builds on earlier work from the authors showing that astrocytic lactate influences the activity of neurons important in these processes. Here they use similar approaches to examine orexin/hypocretin+ (ORX+) neurons in the lateral hypothalamus. There are several strengths, including a multi-platform approach ranging from in vivo to in vitro, transgenic/viral approaches, lactate sensors and patch-clamp electrophysiology. There are, however, some issues that need to be addressed to improve the impact and clarity of the results.

1. In several places results/interpretations reported in the text or legends do not appear to match what is shown in the figures. This in part may be due to the non-standard way of presenting the sleep & wake data (“radar” plots, see below). For example, the *amct4* ko mice are reported in the text to have reductions in wake and nrem sleep time in the first half of the dark phase, but that is not what is shown in Figure 2C: wake as % Rec. time is decreased, and nrem % is increased, which makes sense—as REM % does not change much. Legend in Figure 2, “decreased duration of wakefulness and NREM epochs” is reported. How can a 4-second epoch be decreased in duration? This is confusing. Also, not what I see. I see a reduction in wake and NREM % rec. time and no change in NREM “duration”. This may reflect a problem in how terms are being used and defined (see point 2).
2. There are inconsistencies in how the three main measures of sleep & wake are described (% recording time, duration, and # of events). For example, sometimes the % recording time data are referred to as “duration of time” (line 217), but duration appears to refer to how long an “event” is. BTW, what is an “event”? I assume this is different than a 4 second epoch? If it is a measure of how long an individual occurrence of a state is (more commonly referred to as a “bout”), then the minimum length of a bout needs to be defined. This can range widely from study to study (e.g., 4, 8, 16, 20 seconds). Also, in other places, “episodes” are reported. Is this different than an “event”? Please go through the MS and use consistent terminology (once defined).
3. The radar plots are hard to interpret, especially when lines and symbols overlap and plots are so small. For example, Fig 3c, changes in wake as % rec. time have symbols that are overlapping, yet this is reported as the most significant change in wake (***, although not defined in the legend, I assume 3* are > than 2*). In the same plot, *oMCT2* KO wake duration appears to be at the origin, while the controls are around 6000 seconds—this is not significant? Although I see the advantage of one plot summarizing a lot of data, this is a non-standard way of presenting sleep & wake data. This makes it difficult to compare to earlier work using more common displays. I don’t see much

advantage to doing it this way—and I note the authors used more standard histograms for sleep & wake data in the extended figures. I strongly recommend they use bar histograms with SEMs for the sleep & wake data in the main figures.

Response to comment 1, 2 and 3:

We appreciate the reviewer's feedback. In response, we have replaced the radar plots with bar graphs for a clearer representation of the data. The average duration of an episode is calculated based on consecutive 4-second epochs, allowing the measurement of episodes longer than 4 seconds, with a minimum defined by the 4-second epoch. The percentage of recording time is calculated as the percentage of time spent in each stage (wakefulness, NREM, or REM) over the total considered time. An Episode is defined as an episode spent in wakefulness, we have retained "Episode number" and "Average episode duration (s)" to maintain consistency with our prior work (Clasadonte et al., 2017, Neurons), ensuring clarity for a broader audience compared to the usage of bouts. We have now carefully reviewed the revised manuscript to ensure clarity in terminology.

4. I have several questions about the oMCT2 KO mice data. The representative hypnogram data shown in Figure 3 are internally inconsistent and puzzling. First of all, panel B shows a near complete disorganization of sleep and wake organization and a huge increase in REM sleep in the KO relative to control. This looks like a narcoleptic mouse—which itself could be a very interesting finding, as there is only a trend for a loss of ORX+ neurons from this embryonic KO. Yet, in panel E, another KO mouse shows a distribution of sleep & wake that more closely resembles the control mouse in panel B! While on average, the data may support the author's interpretations, these representative data do not. Also, as a discussion point, why didn't the authors use a viral based approach to KO MCT2 in ORX+ neurons in adults? Isn't it possible that embryonic KO might result in perturbations in ORX+ neurons that go beyond adult cell use of lactate? This might explain why changes in adults differ in some ways from the other manipulations.

In our oMCT2 knockdown mice, we did not observe any narcoleptic events, as defined by the commonly employed criteria (PMID: 19189786; PMID: 10481909). We did not observe in our EEG/EMG analysis any transition from wakefulness into REM nor we did observe any sudden and prolonged muscle tone loss accompanied with increased theta activity during wakefulness. Instead, we noted an increase in both NREM and REM sleep, aligning with existing knowledge that orexin activity (PMID: 35851580) sustains wakefulness. The hypnogram, particularly panel E (now panel G in the revised manuscript), has less NREM and REM compared to oMCT2 in panel B but it is still representative of the phenotype observed.

At the time of our experiment, there were no commercially available viral preparations under the prepro-orexin promoter expressing Cre (precursor of both Orexin-A and Orexin-B). We acknowledge that developmental manipulation could potentially influence behavior. However, even though we observed no decrease in the number of orexin neurons, the re-expression of MCT2 in orexinergic neurons in our oMCT2KO mice demonstrated a rescue of the phenotype. This suggests that there are no developmental impairments underlying the observed sleep/wake impairments. We have addressed it in our Discussion by stating: "Although embryonic manipulation of MCT2 in orexinergic neurons could be considered a confounding strategy, the re-expression of MCT2 during

the adult stage resulted in the recovery of the wakefulness phenotype. This observation suggests the absence of any developmental impairments.”

5. The authors conclude that astrocytic lactate support is a “major regulator” of sleep/wake architecture or is “necessary for maintaining wakefulness during ZT12-18”. These statements are too strong and not supported by the data. The effects are significant, but modest in magnitude and restricted in time. I also wonder if their results might be further restricted to only the first hour or 2 after lights out. If so, this would further circumscribe these results (and raise questions about circadian factors)—but there is no way to know, as hourly values are not shown. Only 6 & 12 hour averages. Moreover, there is no discussion at all about why, if this is a major regulator of sleep & wake, this is restricted to the first half of the dark phase? As a major regulator, one would expect similar effects at all ZT times.

Our results are in line with previous studies employing Orexin knockout mice (mice lacking either orexin expressing neurons or orexin receptors, PMID: 12797957; PMID: 15254084) demonstrating the significance of orexinergic neurons in maintaining wakefulness during the dark phase, while their impact during the light phase is minimal to none. In the Supplementary figures provided in this revised version (Fig. S1, S3 and S5) it is further clear that manipulation does not affect the light phase and has a major impact in the first part of the dark phase.

6. The authors should really address caveats in the interpretation of their results—especially as the discussion is too long; it should be shortened to accommodate a discussion of caveats. The different manipulations produce similar, but also different results (e.g, 4-CIN vs. MCT4 KO vs. oMCT2 KO), and this is likely due to the strengths and weaknesses of each approach. For example, the minipump infusions are into the LH, which contain neurons other than ORX+. So, how far is the effective infusion zone? What other neurons are there that can directly or indirectly impact sleep & wake architecture? Do these manipulations impact neurons that regulate core temperature, motor activity or pain? See above for points regarding the oMCT2 KO mice. We thank the reviewer for the suggestion. We have now added part of these limitations of our approach into the Discussion section by stating: “We also detected an effect during the light phase, and it is noteworthy that we cannot exclude off-target effects of 4-CIN in this experiment, including extracellular and intracellular acidification due to impaired proton transport, inhibition of mitochondrial pyruvate transport, or effects on other cell types, rather than astrocytes expressing MCTs.” and “In line with previous evidence demonstrating fragmented wakefulness during the dark phase in orexin-deficient mouse models (Mochizuki, 2004; Willie, 2003), we did not observe any phenotype in the light phase in aMCT4 KD and oMCT2 KD mice. This finding confirms the role of orexinergic neurons and suggests that other neurons in the same area, such as melanin-concentrating hormone neurons (MCH), may not be dependent on astrocytic lactate. In addition, we also identified a consistent role for lactate as a fuel during the first 6 hours of the dark phase. The lack of effects in the latter part (ZT18-24) may indicate reduced orexinergic activity, potentially sustained by glucose or other metabolites.”

Additional points

1. male and female mice were used. Were there any sex differences?

We did not observe any sex differences and mice were pooled.

2. For the lactate sensor data, only EEG signals were used to score REM, Wake and NREM? How was this validated? cortical EEG signals are not the best means of scoring wake vs REM.

EEG was paired with EMG recordings (now added in Fig.5a)

3. Did any of these manipulations effect core temperature?

In this manuscript, we did not investigate core temperature effects in our KD mice. Based on the evidence in the literature orexinergic neurons appear to have a role in the control of core temperature only under stress (fasting:PMID: 29426934 or exercise:PMID: 31804545, PMID: 36963505), therefore we do not expect changes in core temperature in this study.

Reviewer #5 (Remarks to the Author):

REVIEWER COMMENTS

Reviewer #1 (Remarks to the Author):

The authors have addressed my concerns and I'm almost completely satisfied with the revised version.

I've just noticed that the reverse KO primer should be in black in the legend to the schematic Supplementary Fig. 2a.

Reviewer #2 (Remarks to the Author):

The authors have sufficiently addressed the majority of this reviewer's concerns, including substantial revisions to text and additional experiments performed.

One concern remains (line 585) where the authors state that orexin neurons express Kir6.1 and Sur1. First, this manuscript does not demonstrate this and relies on one previously published paper that only looks at Kir6.x subunits, not Sur1 or Sur2A/B. Therefore, no claim on SUR1 v SUR2 can be made. Moreover, this paper uses 1 antibody for Kir6.1 and 1 for Kir6.2, only 1 field of view is shown in the paper, and there is no quantification. There are numerous other papers that demonstrate Kir6.2-KATP channels are found in the hypothalamus as well. If the authors do not wish to quantify Kir6.1 v Kir6.2 expression, then this statement should be modified to state, "Finally, orexin neurons express functional KATP channels 21,33,52, 20 known to modulate their activity." Since antibodies for KATP channel subunits are known to cross react, it is important to do proper controlled experiments while exploring expression. Or use other methods (e.g. RNA-scope) to corroborate historical findings.

Once this concern is addressed, this paper is now suitable for publication and is of broad interest to the field.

Reviewer #3 (Remarks to the Author):

I am satisfied with the revised version

Reviewer #4 (Remarks to the Author):

The authors have been mostly responsive to my concerns. I have some remaining concerns that were not addressed and/or raise questions about the methodology.

Major issues

My original comment: "The authors conclude that astrocytic lactate support is a major regulator' of sleep/wake architecture or is 'necessary for maintaining wakefulness during ZT12-18'. These statements are too strong and not supported by the data. The effects are significant, but modest in magnitude and restricted in time. I also wonder if their results might be further restricted to only the first hour or 2 after lights out. If so, this would further circumscribe these results (and raise questions about circadian factors)—but there is no way to know, as hourly values are not shown. Only 6 & 12 hour averages. Moreover, there is no discussion at all about why, if this is a major regulator of sleep & wake, this is restricted to the first half of the dark phase? As a major regulator, one would expect similar effects at all ZT times."

The authors' response does not completely address these concerns: "Our results are in line with previous studies employing Orexin knockout mice (mice lacking either orexin expressing neurons or orexin receptors, PMID: 12797957; PMID: 15254084) demonstrating the significance of orexinergic neurons in maintaining wakefulness during the dark phase, while their impact during the light phase is minimal to none. In the Supplementary figures provided in this revised version (Fig. S1,S3 and S5) it is further clear that manipulation does not affect the light phase and has a

major impact in the first part of the dark phase.

The authors may have tempered some of the original language, but unfortunately still overstate the significance of the results of the MCT4 mutants. There is no convincing evidence that astrocytic lactate support via MCT4 is, as now said, is “necessary for maintaining prolonged wakefulness”, or “... that astrocytic MCT4... is necessary for consolidating wakefulness during the dark phase”. The effects based on the MCT4 mice are small, and the new supplementary figure indicates that this (maybe) is indeed restricted to a single hour of the dark phase based on % time awake, not episode duration—which is the best metric of ‘consolidation’ (new supplementary figure 3). This is not evidence of a “major impact in the first part of the dark phase”. What I see is that MCT4 maybe contributes to waking time as a % at a single hour, but this does not support such a strong statement. Even this conclusion is questionable as the results of the ANOVA in these analyses (meta data, statistics table, for supplementary figure 3) show no significant effects in the ANOVA (also, why are the main effects not reported? Minimally genotype as a level). Post-hoc tests are only warranted if there is significance in the (repeated) ANOVA, so why then did they proceed with post-hoc tests, and base their interpretation of these results on them? This is extremely troubling. The proper interpretation is that there is only a trend towards significance—given the number of mice per group, I doubt this would resolve with more mice, but a power assessment seems needed.

On a related point, I don’t see a justification for dividing the data into 6 hour chunks for the dark phase as is done in this paper. I may have missed this in the methods, but if I am correct this rationale should be provided; referring to a prior study that did the same thing, without justification would not be responsive.

Additional points:

The authors now present their data in more standard ways—and this improves the presentation. Unfortunately, in several cases, they do not adjust their Y axes in scale, so in some cases, the data is piled up at the bottom of the graph and is largely uninterpretable. A pile of bubbles with a line over them. I strongly recommend, in the interests of transparency and good presentation, that they take another look at some of the figures and rescale where needed so that their readers can see the data.

Reviewer #5 (Remarks to the Author):

Reviewer #1 (Remarks to the Author):

The authors have addressed my concerns and I'm almost completely satisfied with the revised version. I've just noticed that the reverse KO primer should be in black in the legend to the schematic Supplementary Fig. 2a.

We thank the reviewer for the suggestion. We have now changed the color.

Reviewer #2 (Remarks to the Author):

The authors have sufficiently addressed the majority of this reviewer's concerns, including substantial revisions to text and additional experiments performed.

One concern remains (line 585) where the authors state that orexin neurons express Kir6.1 and Sur1. First, this manuscript does not demonstrate this and relies on one previously published paper that only looks at Kir6.x subunits, not Sur1 or Sur2A/B. Therefore, no claim on SUR1 v SUR2 can be made. Moreover, this paper uses 1 antibody for Kir6.1 and 1 for Kir6.2, only 1 field of view is shown in the paper, and there is no quantification. There are numerous other papers that demonstrate Kir6.2-KATP channels are found in the hypothalamus as well. If the authors do not wish to quantify Kir6.1 v Kir6.2 expression, then this statement should be modified to state, "Finally, orexin neurons express functional KATP channels 21,33,52, 20 known to modulate their activity." Since antibodies for KATP channel subunits are known to cross react, it is important to do proper controlled experiments while exploring expression. Or use other methods (e.g. RNA-scope) to corroborate historical findings.

Once this concern is addressed, this paper is now suitable for publication and is of broad interest to the field.

We understand these concerns and the importance of validating these previous findings with proper controlled experiments, so we changed that statement with: "Finally, orexin neurons express functional KATP channels 21,33,52, 20 known to modulate their activity." As suggested by this reviewer.

Reviewer #3 (Remarks to the Author):

I am satisfied with the revised version

Reviewer #4 (Remarks to the Author):

The authors have been mostly responsive to my concerns. I have some remaining concerns that were not addressed and/or raise questions about the methodology.

Major issues

My original comment: “The authors conclude that astrocytic lactate support is a major regulator’ of sleep/wake architecture or is ‘necessary for maintaining wakefulness during ZT12-18’. These statements are too strong and not supported by the data. The effects are significant, but modest in magnitude and restricted in time. I also wonder if their results might be further restricted to only the first hour or 2 after lights out. If so, this would further circumscribe these results (and raise questions about circadian factors)—but there is no way to know, as hourly values are not shown. Only 6 & 12 hour averages. Moreover, there is no discussion at all about why, if this is a major regulator of sleep & wake, this is restricted to the first half of the dark phase? As a major regulator, one would expect similar effects at all ZT times.”

The authors’ response does not completely address these concerns: “Our results are in line with previous studies employing Orexin knockout mice (mice lacking either orexin expressing neurons or orexin receptors, PMID: 12797957; PMID: 15254084) demonstrating the significance of orexinergic neurons in maintaining wakefulness during the dark phase, while their impact during the light phase is minimal to none. In the Supplementary figures provided in this revised version (Fig. S1,S3 and S5) it is further clear that manipulation does not affect the light phase and has a major impact in the first part of the dark phase.

We have addressed the reviewer’s new concerns in the section below.

The authors may have tempered some of the original language, but unfortunately still overstate the significance of the results of the MCT4 mutants. There is no convincing evidence that astrocytic lactate support via MCT4 is, as now said, is “necessary for maintaining prolonged wakefulness”, or “... that astrocytic MCT4... is necessary for consolidating wakefulness during the dark phase”. The effects based on the MCT4 mice are small, and the new supplementary figure indicates that this (maybe) is indeed restricted to a single hour of the dark phase based on % time awake, not episode duration—which is the best metric of ‘consolidation’ (new supplementary figure 3). This is not evidence of a “major impact in the first part of the dark phase”. What I see is that MCT4 maybe contributes to waking time as a % at a single hour, but this does not support such a strong statement. Even this conclusion is questionable as the results of the ANOVA in these analyses (meta data, statistics table, for supplementary figure 3) show no significant effects in the ANOVA (also, why are the main effects not reported? Minimally genotype as a level). Post-hoc tests are only warranted if there is significance in the (repeated) ANOVA, so why then did they proceed with post-hoc tests, and base their interpretation of these results on them? This is extremely troubling. The proper interpretation is that there is only a trend towards significance—given the number of mice per group, I doubt this would resolve with more mice, but a power assessment seems needed.

Reviewer raises some important points about the strength of our conclusion; the use and interpretation of our statistics, in particular two-way ANOVA, and finally the division of the dark phase into a 6-hour time block between ZT 12 and ZT18.

1 and 2- We acknowledge our oversight in not reporting all main effects from the 2-way ANOVA statistics performed in the Extended Figures (24h analysis of % WAKE, %NREM and %REM, 1h bin). We have rectified this, revealing a significant difference between genotypes in the aMCT4 KO group compared to the control group ($p=0.0185$ in WAKE and $p=0.0306$ in NREM). These results justified our decision to conduct subsequent post hoc tests.

As suggested by the reviewer, consolidated wakefulness is determined by the length of the wakefulness episode; aMCT4 KO mice exhibited a significant reduction in the duration of wakefulness in the first 6 hours of the dark ZT12-18 (Fig. 2e). (Please see the below response for justification of the ZT12-18 and ZT18-24 hours division). However, we agree with the reviewer and have softened the text as follows: “Together, these results suggest that astrocytic MCT4 **supports** wakefulness during ZT12-18 and consistent with the transport of lactate into the extracellular milieu by MCT4 given that lactate can rescue the aMCT4 KD phenotype.” Lines 115 and 248

“Here, we show that astrocytic MCT4, but not MCT1, **supports** prolonged wakefulness.” line 495

On a related point, I don't see a justification for dividing the data into 6-hour chunks for the dark phase as is done in this paper. I may have missed this in the methods, but if I am correct this rationale should be provided; referring to a prior study that did the same thing, without justification would not be responsive.

3- We decided to split the dark phase into two periods (ZT12-18 and ZT18-24) based on previously published observations demonstrating that mice are more active and awake during the first 6 hours of the dark phase (PMID: 28456660; PMID: 29581380; PMID: 24358130; PMID: 31611779; PMID: 26863349; PMID: 22578011; PMID: 12749544; PMID: 28548639; PMID: 33903668; PMID: 30354930; PMID: 28867552; PMID: 28167901; PMID: 19186164). These observations have been validated in this study. In fact, we observed that mice are awake about 80% of the time during ZT12-18 and about 60% during ZT18-24, as shown in Figures 1c, 2c, and 3c. Moreover, in line with the role of orexinergic neurons in sustaining wakefulness during the dark phase, we decided to adopt the 6h division during the dark.

We have clarified this methodological decision in the Materials and Methods section as follows: “Dark phase was divided into ZT12-18 and ZT18-24 based on prior observations (PMID: 28867552; PMID: 28167901; PMID: 19186164) and studies showing differential activity in the dark phase (PMID: 28456660; PMID: 29581380; PMID: 24358130).

Additional points:

The authors now present their data in more standard ways—and this improves the presentation. Unfortunately, in several cases, they do not adjust their Y axes in scale, so in some cases, the data is piled up at the bottom of the graph and is largely uninterpretable. A pile of bubbles with a line over them. I strongly recommend, in the interests of transparency and good presentation, that they take another look

at some of the figures and rescale where needed so that their readers can see the data.

As per the request from the reviewers, we previously changed the radar plots presentation to histograms. The radar plots allowed us to show REM with different scales than NREM and Wake. However, with the standard histogram approach that has been requested the small duration of REM compared to wake and NREM makes it more difficult to see the individual data points during REM sleep. In this revised version we have adjusted the Y-scales as much as possible, but due to the need to display every value accurately, the Y-axis could not be further adjusted. Source data are also provided in this paper for transparency.

Reviewer #5 (Remarks to the Author):

REVIEWERS' COMMENTS

Reviewer #4 (Remarks to the Author):

The authors have been responsive to review. This revised version is a vastly improved study that is now more accessible to the interested reader.